

# Multivariate Anomaly Detection for Earth Observations: A Comparison of Algorithms and Feature Extraction Techniques

Milan Flach[1], Fabian Gans[1], Alexander Brenning[2,4], Joachim Denzler[3,4,5], Markus Reichstein[1,4,5], Erik Rodner[3,4], Sebastian Bathiany[6], Paul Bodesheim[1], Yanira Guanche[3,4], Sebastian Sippel[1], and Miguel D. Mahecha[1,4,5]

[1]Max Planck Institute for Biogeochemistry, Department Biogeochemical Integration, P. O. Box 10 01 64, D-07701 Jena, Germany
[2]Friedrich Schiller University Jena, Department of Geography, Jena, Germany
[3]Friedrich Schiller University of Jena, Department of Mathematics and Computer Sciences, Computer Vision Group, Jena, Germany
[4]Michael Stifel Center Jena for Data-driven and Simulation Science, Jena, Germany
[5]German Centre for Integrative Biodiversity Research (iDiv), Leipzig, Germany
[6]Wageningen University, Department of Environmental Sciences, Wageningen, Netherlands

*Correspondence to:* Milan Flach (milan.flach@bgc-jena.mpg.de)

**Abstract.** Today, many processes at the Earth's surface are constantly monitored by multiple data streams. These observations have become central to advance our understanding of e.g. vegetation dynamics in response to climate or land use change. Another set of important applications is monitoring effects of climatic extreme events, other disturbances such as fires, or abrupt land transitions. One important methodological question is how to reliably detect anomalies in an automated and generic way
within multivariate data streams, which typically vary seasonally and are interconnected across variables. Although many algorithms have been proposed for detecting anomalies in multivariate data, only few have been investigated in the context of Earth system science applications. In this study, we systematically combine and compare feature extraction and anomaly detection algorithms for detecting anomalous events. Our aim is to identify suitable workflows for automatically detecting anomalous patterns in multivariate Earth system data streams. We rely on artificial data that mimic typical properties and anomalies in
multivariate spatiotemporal Earth observations. This artificial experiment is needed as there is no 'gold standard' for the identification of anomalies in real Earth observations. Our results show that a well chosen feature extraction step (e.g. subtracting seasonal cycles, or dimensionality reduction) is more important than the choice of a particular anomaly detection algorithm. Nevertheless, we identify 3 detection algorithms ($k$-nearest neighbours mean distance, kernel density estimation, a recurrence approach) and their combinations (ensembles) that outperform other multivariate approaches as well as univariate extreme
event detection methods. Our results therefore provide an effective workflow to automatically detect anomalies in Earth system science data.

**Keywords.** statistical process control, process monitoring, Earth observations, artificial data, multivariate outlier detection, novelty detection, detection of extreme events, anomaly detection, event detection, k-nearest neighbours, kernel density estimation, recurrences, support vector data description, kernel null foley-sammon transform, mahalanobis distance, Hotelling's
$T^2$, multivariate exponential moving average.



# 1  Introduction

The Earth system can be conceptualized as a system of highly interconnected subsystems (e.g. atmosphere, biosphere, hydrosphere, lithosphere). Each of these subsystems can be monitored and characterized by multiple variables. Technological progress over the past decades has led to a boost in satellite technologies (Pfeifer et al., 2011; Nagendra et al., 2012) as well as ground station development and routine monitoring (Baldocchi et al., 2001; Dorigo et al., 2011; Ciais et al., 2014). Additionally, advanced computational methods efficiently integrate remote sensing and in-situ information to routinely derive novel data products (e.g., Beer et al., 2010; Jung et al., 2011; Tramontana et al., 2016). One key scientific challenge is co-interpreting these multiple views on the Earth system, in particular to address the impacts of changes in the climate system, the land use system, and other transformations.

Of particular importance is the analysis of extreme events like droughts, fires, heat waves or floods which are expected to change in a future climate (Kharin et al., 2013). One matter of concern are changes in hydrometeorological extremes that may translate into anomalies in vegetation dynamics, or extremes in vegetation dynamics that might result from slight changes in climatological conditions or human intervention, and that can have severe consequences for vegetation and the carbon cycle (Easterling et al., 2000; Meehl and Tebaldi, 2004; Seneviratne et al., 2012; Reichstein et al., 2013). Apart from natural events, one also aims at detecting events that are a direct consequence of human interference, e.g., detecting deforestation activities is required to assess the compliance with laws or agreements on forest conservation and climate change.

The flood of observational data is accompanied by a similar increase in data from Earth system models (Overpeck et al., 2011). As large amounts of data are difficult to handle and to translate to the quantities of human interest, it can be easy to overlook events of particular importance. For example, using a simple semi-automatic detection scheme to identify abrupt climate shifts in simulations of future climate, Drijfhout et al. (2015) found a number of abrupt events that have previously been overlooked in simulations.

In observations, anomalous events are often detected using extreme event detection methods suitable for univariate data streams (e.g., Alexander et al., 2006; Rahmstorf and Coumou, 2011; Zhou et al., 2011; Donat et al., 2013; Lehmann et al., 2015). Univariate extreme event detection can also be used to infer knowledge about underlying drivers of extremes (Zscheischler et al., 2014a); it is particularly valid when the variable of interest is either of specific importance or integrates a wide array of relevant processes. However, some information might only be inferred when taking the multivariate combination of several data streams into account (Vicente-Serrano et al., 2010; Seneviratne et al., 2012; Fischer, 2013; Zscheischler et al., 2015). For instance, a significant fraction of carbon extremes events in Europe is not associated with univariate climate extremes (Zscheischler et al., 2014b). Earth observations are multivariate and naturally characterized by strong dependencies and correlations in space, time, and across dimensions (Leonard et al., 2013). We assume that any suitable anomaly detection algorithm needs to consider these data properties. By considering multivariate constellations for anomaly detection, it might become possible to gain further information, i.e. about anomalies which cannot be detected with univariate extreme event detection methods (for a review of approaches see, e.g., Ghil et al., 2011).





Multivariate approaches in geoscience make use of anomalies occurring simultaneously in multiple data streams, often referred to as coincidences or coexceedances (e.g., Donges et al., 2011b; Rammig et al., 2015; Zscheischler et al., 2015; Donges et al., 2016; Guanche et al., 2016). An alternative is the copula approach introduced to the field e.g. by Schoelzel and Friedrichs (2008); Durante and Salvadori (2010). However, the copula approach so far is limited to 2 or 3 simultaneous data streams (Mikosch, 2006) which makes it unsuitable for high dimensional data as used in this paper.

Interestingly, there are multiple industrial applications that likewise require anomaly detection. In this context, anomaly detection has become a standard procedure in the wake of Harold Hotelling's publication of the $T^2$ control chart in 1947 (Hotelling, 1947; Lowry and Woodall, 1992). Consider, for instance several sensors observing some industrial production chain. These (potentially correlated) sensor data streams can be monitored with a Statistical Process Control (SPC) algorithm (Lim et al., 2014; Ge et al., 2013; Lowry and Montgomery, 1995). The basic idea is to raise an alarm as soon as an anomaly according to the SPC is detected, meaning that the production chain is 'out of control'. Despite the obvious analogy, the ideas of SPC are largely unknown in the geoscience community to the best of our knowledge. Conceptually, the industrial application is equivalent to the idea of monitoring environmental variables. However, data differ. Earth observations (EOs) exhibit strong (potentially non-linear) dependencies among the variables, seasonal cycles are typically present in both temporal mean and variance. The variables may also encode dynamic feedbacks and abrupt transitions. EOs are possibly more strongly corrupted by noise compared to industrial applications. Furthermore industrial applications are typically less affected by low-frequency variability than Earth observations. The most problematic aspect when considering SPC concepts in Earth system sciences is, however, defining states of normality.

The objective of this study is to provide an overview and comparison of anomaly detection algorithms and their combination with feature extraction techniques for identifying multivariate anomalies in EOs. Therefore we define an anomaly to be any consecutive spatiotemporal part of the Earth system data cube (data cube), which differs with respect to the mean, the variance, the amplitude of the seasonal cycle of trends from the 'normal' rest of the data cube. We adapt algorithms from SPC and novelty detection. The study is structured as follows: First, we create a series of artificial Earth system data cubes that try to mimic a series of real world features (in terms of multiple variables, seasonal cycles and correlation structure, etc). We are aware that these artificial data cubes are not 'real' simulations of Earth system data cubes. However, relying on artificial data in this paper is motivated by the fact that a meaningful quantitative evaluation of unsupervised anomaly detection algorithms and feature extraction techniques in 'real' Earth observation data is difficult due to the lack of ground-truth data (Zimek et al., 2012). Second, we use these artificial data to evaluate the capability of different algorithms to detect multivariate anomalous events. Specifically, we evaluate the performance of the algorithms to detect multivariate changes in the mean (comparable to an extreme event), the amplitude of the annual cycle, the variance and onset of trends. Using the artificial dataset as testbed we apply various feature extraction schemes (Sect. 3.1), several detection algorithms (Sect. 3.2) as well as combinations of detection algorithms (ensembles, Sect. 3.4) to compare their performance in identifying anomalous events (Sect. 3.3). From this comparison we select suitable combinations of feature extraction (Sect. 4.1) and a few algorithms (Sect. 4.2) as well as ensembles of algorithms (Sect. 4.3) as the best ones applicable to EOs including suggestions for their specific usage (Sect. 5).




## 2 Experimental Setup

### 2.1 Generation Principle of the Artificial Data

Ground truth for detecting anomalies in multivariate data is rare, in particular for detecting anomalies in 'real' EOs. Thus, we generate artificial data that represent common properties of EOs, including anomalies. In particular, we focus on the existence

5 of seasonality, correlations among variables, and non-Gaussian distributions. Data generation assumes that each subsystem of the Earth has uncorrelated intrinsic properties, i.e. it is dominated by a few independent components. Consequently, generating these independent components (which cannot directly be monitored) is the first step. We then derive variables that contain elements of all independent components and correspond to the 'observable' measurements as a set of correlated variables (Fig. 1).

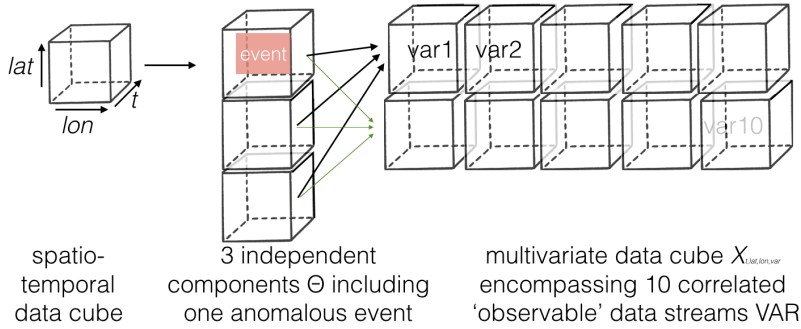

spatio-
temporal
data cube

independent
components $\Theta$ including
one anomalous event

multivariate data cube $X_{t,lat,lon,var}$
encompassing 10 correlated
'observable' data streams VAR

**Figure 1.** Combination of 3 independent component cubes to derive 10 correlated variables $X$ as 'observable' measurements. The anomalous event is propagated into some variables of $X$.

More precisely, as basic version we create 3 independent components for the artificial data, each consisting of a signal (Gaussian, $sd = 1.0$) which includes seasonality in some cases (Sect. 2.3). Anomalous events are induced in one of the independent components for which we track the exact spatiotemporal location. These 3 independent components are then weighted with randomly generated linear (or non-linear, Sect. 2.3) weights to create a set of 10 correlated variables, which represent the artificial data cube, i.e. try to mimic 'observable measurements'. We add some additional measurement noise (Gaussian,

$sd = 0.3$) to the data cube. For more technical details of this generation scheme we refer the reader to the Appendix B.

Our standard data cube $X_{t_{i,j},lat,lon,var}$ encompasses $t_{i,j} = 1, \ldots, T$ time steps ($T = 300$) corresponding e.g. to a 6.5-year time series of satellite images in 8-day intervals, $lat = 1, \ldots, LAT$ latitudes ($LAT = 50$), $lon = 1, \ldots, LON$ longitudes ($LON = 50$) and $var = 1, \ldots VAR$ data streams, or variables ($VAR = 10$).

### 2.2 Generating Anomalous Events

Anomalous events are introduced on the independent components only and then propagated from the independent component to some of the variables in the data cube with random weights. The anomalies are contiguous in space and time. The center





of the anomaly is assigned randomly. The challenge is to detect the propagated anomaly through the unsupervised algorithms, i.e. without using the information about the spatiotemporal location of the anomaly. With this data cube generation scheme, we can generate anomalies by controlling the type of the anomalous event (event type), the magnitude of the anomalous event as well as the spatiotemporal location.

We create 4 data cubes using the following temporary event types:

a  Shift in the baseline, i.e. shift of the running mean of a time series (*BaseShift*) (Fig. 2 a)

b  Onset of a trend in the time series (*TrendOnset*) (Fig. 2 b)

c  Change in the amplitude of the mean seasonal cycle of a time series (*MSCChange*) (Fig. 2 c)

d  Change in the variance of the time series (*VarianceChange*) (Fig. 2 d)

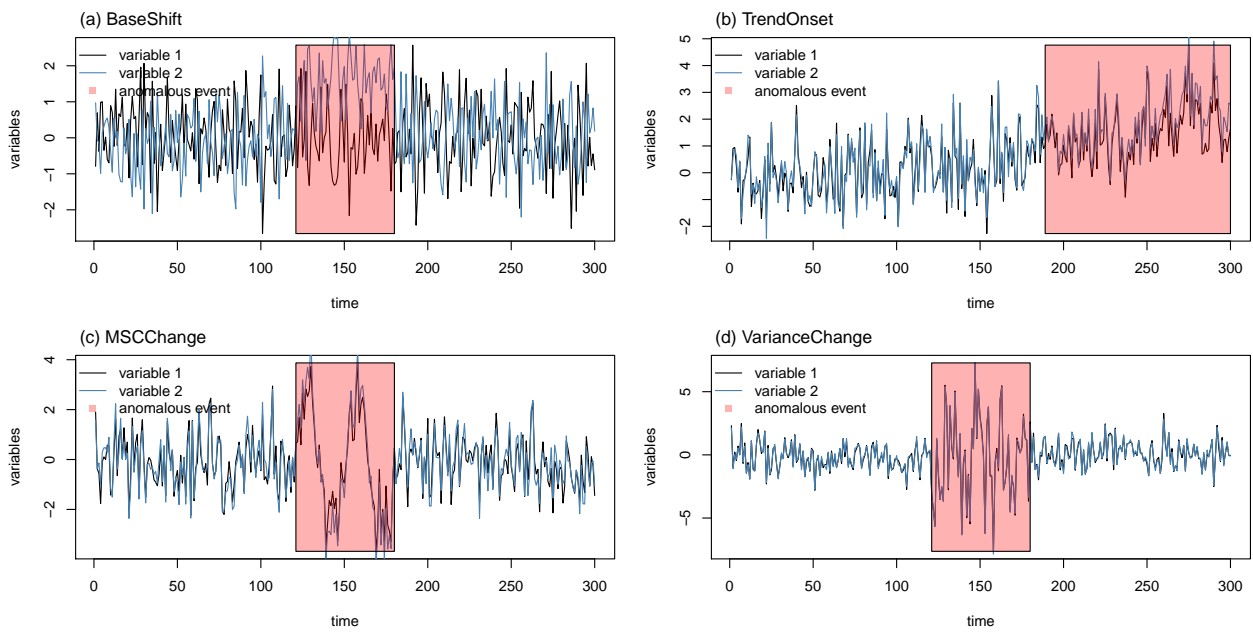

**Figure 2.** Visualization of the 4 different event types (a-d) with 2 variables along time. The 2 variables contain an anomalous event (here: 60 time steps long) which is propagated through the underlying independent components with randomly drawn weights within the generation process of the variables.

**2.3  Additional Complications**

Apart from the basic data cubes we want to test the influence of a certain complication on the anomaly detection algorithm. In order to do so, we create data cubes, each with one added complication, i.e. we increase the number of independent components (*MoreIndepComponents*) or use a squared dependency among independent components (*NonLinearDep*) instead of a linear one. Furthermore typical EOs are often driven by extrinsic forcings, i.e. the Earth's solar system orbit, rotation, and axis tilt, thus

we add a seasonal cycle modifying the signal (*SeasonalCycle*). In a global context, the mean is rarely constant; we therefore



introduce a linear latitudinal trend into the baseline (*LatitudinalGradient*). In the basic case, the signal of our independent components follows a Gaussian distribution. In the more complicated versions, we also implement alternative scenarios with Laplacian ('doubly exponential') distributed signals (*LaplacianNoise*) and signals which exhibits spatiotemporal correlation with red noise (*CorrelatedNoise*). Signal-to-noise ratio is 0.3 in the basic version, one complication increases the signal-to-noise

ratio to 1.0 (*NoiseIncrease*). Also the shape and duration of anomalous events differs. We double (*LongExtremes*) or reduce the temporal duration of the anomalous events (*ShortExtremes*) and change the spatial shape from rectangular to randomly affecting neighbouring grid cells (*RandomWalkExtreme*).

## 2.4 Experiment design

Each data cube with a specific type of the event is generated 20 times, each time with a different magnitude of the anomalous

event (Appendix B). We introduce 10 spatially contiguous anomalous events into the independent components, with a spatial extent of 20 latitude and longitude steps each. Each event has a temporal extent of 5 time steps (which would be equivalent to 40 consecutive anomalous days in a 6.5 year record). Our total amount of anomalies equals about 3 % of the total data cube which we consider to be a realistic scenario (comparable to e.g., Zscheischler et al., 2014a). Some latitudes and longitudes do not exhibit any anomaly by design. The algorithms (Sect. 3.2) are expected to be able to deal with parts of the data cube that

do not exhibit anomalies at all, as this is also very likely to happen for applications in real Earth observations.

Our experiment comprises 36 different event type combinations of complications, each repeated 20 times with varying event magnitudes (Appendix B). The entire set of artificial data cube consists of 720 data cubes, corresponding to ≈87 GB of data [1].

## 3 Workflows to Detect Anomalies

The idea of this study is to elaborate workflows that contain both data preprocessing via feature extraction and algorithms

for the detection of anomalous events (Fig. 3). In the following we introduce these 2 elements separately and explain the performance evaluation strategy afterwards.

### 3.1 Feature extraction

'Feature extraction' is a process to derive information from the data and condense it into non-redundant characteristic patterns. This may facilitate data interpretation (van der Maaten, 2009). In our study the aim is to maximize the event detection rate

by providing relevant features. Feature extraction is often an element of data preprocessing. A very simple form could be to subtract the mean seasonal cycle (the anomaly time series becomes the feature then). Here, we concentrated mainly on feature extraction methods that are used in the context of classical multivariate SPC (Lowry and Montgomery, 1995), data-based process monitoring in industry (Ge et al., 2013), and univariate extreme event detection. The following feature extraction methods are used in this study:

---

[1]Code to reproduce the data farm is provided in Appendix B.



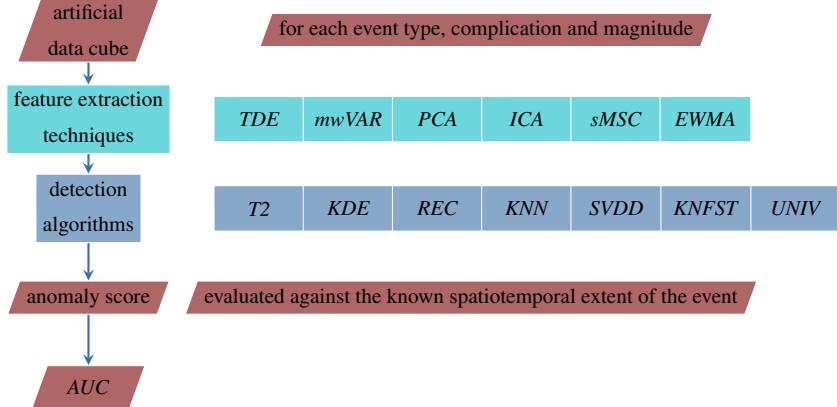

**Figure 3.** Data processing for detecting multivariate anomalies. We extract relevant features from each artificial data cube before applying the detection algorithms. The detection algorithms output some anomaly score which we evaluate against the known extent of the event using the Area Under the Curve (*AUC*). Feature extraction elements on the right hand side are understood as options and can be combined with each other.

Subtracting the Median Seasonal Cycle (*sMSC*) is one way to deseasonalize time series. Deseasonalization may be instrumental in detecting anomalous events across different seasons. The remaining part of the time series is often referred to as anomalies and used here as input feature.

Variance in Moving Window (*mwVAR*) is a popular technique for detecting trends in the variance in univariate time series
(e.g., Huntingford et al., 2013). We compute the *mwVAR* with a window size of 10 and subtract the median *mwVAR*. We use the estimates of the temporal moving window variance as feature to detect multivariate anomalies in the variance.

Time Delay Embedding (*TDE*) increases the feature vector $Y_t$ with time delayed vectors ($Y_t = (X_t - 0\tau, X_t - 1\tau, X_t - (m-1)\tau)$) to include temporal context information. In the univariate case, this approach ideally creates an image of the attractor of a dynamical system (Takens, 1981). The theoretical consideration does not hold true for high dimensional multivariate data,
but is nevertheless used in practical applications to include information of the dynamics in the feature vector (e.g., Koçak et al., 2004; Ge et al., 2013; Smets et al., 2009). Critical hyperparameters are the time delay $\tau$ and the number of dimensions $m$. We fix $m$ to 3 and $\tau$ to 6 which is a compromise between the typical choice of the first zero crossing of the temporal autocorrelation function (Webber and Marwan, 2015) (here: 11.5 corresponding to one quarter of the annual cycle with 46 time steps) and an accurate temporal detection (requires small $\tau$).

Principal Component Analysis (*PCA*) is a data rotation, used to find an orthogonal (uncorrelated) subspace of the data of $n_{PC} \leq VAR$ variables (Von Storch and Zwiers, 2001). We choose $n_{PC}$ such that at least 95 % of the variance in the original data cube are explained. By assuming a homogeneous covariance structure within the entire data cube, we perform the *PCA* globally, i.e. with the same rotation matrix for all grid cells. The combination of *TDE* and *PCA* is sometimes referred to as dynamic *PCA* when considering subsequent lags in the time series (Lee et al., 2004).



Independent Component Analysis (*ICA*) can be regarded as a nonlinear alternative to PCA; it has become a standard technique of data-based process monitoring, trying to separate different sources of data by maximizing the negentropy, a measure of non-Gaussianity of the data (Hyväringen and Oja, 2000)[2]. We apply *ICA* globally to each data cube. The hyperparameter is the number of independent components (sources). We choose the number of independent components to be equal to $n_{PC}$ (see *PCA*) for consistency reasons (Majeed and Avison, 2014).

Exponentially Weighted Moving Average (*EWMA*) is one way of reducing the noise of the time series and taking temporal information into account. It is common in the context of classical multivariate SPC to detect only 'significant' outliers (Lowry and Woodall, 1992). The multivariate feature time series Y is computed recursively as

$$Y_{t_i} = \lambda X_{t_i} + (1 - \lambda) Y_{t_{i-1}}. \tag{1}$$

The hyperparameter $\lambda$ determines the degree of exponential weighting between 1 (no weighting) and zero (common choice $0.1 \leq \lambda \leq 0.3$, Santos-Fernández, 2012). We stay in this range with $\lambda = 0.15$.

There is of course a multitude of alternative approaches available in literature, but we focus on the previously summarized ones as they are widely used and efficiently implemented. Furthermore, different feature extraction methods can also be combined (Fig. 3). As the number of possible combinations is considerably large, we focus here on dimensionality reduction techniques (*ICA*, *PCA*) combined with some *EWMA* to reduce the noise level afterwards. Depending on the event type and complication, additionally removing seasonality (*sMSC*) or including the variance *mwVAR* seems to be straightforward. Information about the dynamics (*TDE*) can be included before applying dimensionality reduction techniques, to keep the dimensionality of the system as low as possible. In the following, combinations are noted in the order in which they were applied (e.g., *PCA_EWMA* means first applying *PCA*, then applying *EWMA* on the *PCA* features).

## 3.2 Anomaly Detection Algorithms

We use several detection algorithms which we implemented in the julia package MultivariateAnomalies.jl[3]. We fix model parameters for the entire data cube. Model parameters ($\sigma, \varepsilon, Q, \mu$) and the models themselves (Support Vector Data Description, Kernel Null Foley-Sammon Transform, see below) are estimated on a random subsample of 5000 data points obtained from the entire data cube. To account for variability of the model parameter estimation, we resample 3 times. More resampling is affordable due to high computational costs of processing the large number of data cubes. However, very little random variability is observed with this sample size for the best algorithms. Thus, we consider a resampling of 3 to be sufficient for a first attempt accounting for variability in the parameterization. The following algorithms are investigated for anomaly detection.

Univariate Approach (*UNIV*). A simple approach to define extremes in univariate data is to identify all points above (or below) a certain quantile. This so-called 'peak-over threshold' approach can be transferred to deal with multiple univariate data streams. In this case, one would consider a data point to be extreme, if one or several of the univariate variables are below or above a certain quantile threshold (here: globally) (e.g., Ledford and Tawn, 1996; Bae et al., 2003; Donges et al., 2016).

---

[2] We use the *fastICA* algorithm implemented in the julia package MultivariateStats.jl (https://github.com/JuliaStats/MultivariateStats.jl).

[3] https://github.com/milanflach/MultivariateAnomalies.jl



Applications of the so-called cooccurrence or coincidence analysis can be found in Donges et al. (2011b); Rammig et al. (2015); Zscheischler et al. (2015); Guanche et al. (2016). For comparing the algorithms, we are interested in the information that at least one variable is above a certain threshold. We compute this information for all possible thresholds to get a score, i.e. a ranking of the extremeness of the data points.

Hotelling's $T^2$ (*T2*) computes the squared Mahalanobis distance of each data point $X_t$ to its temporal mean $\mu$ weighted with the covariance matrix $Q$ (Hotelling, 1947):

$$(X_t - \mu)' Q^{-1} (X_t - \mu) \tag{2}$$

A crucial prerequisite is the estimation of the covariance matrix $Q$, which is estimated from the random subsample of 5000 data points. Combining the feature extraction *EWMA* with *T2* equals the traditional multivariate exponential weighted moving

average (Lowry and Woodall, 1992; Lowry and Montgomery, 1995).

    Apart from computing weighted distances to the mean (like *T2*), it is also possible to compute pairwise Euclidean distances in variable space $d(X_{t_i}, X_{t_j})$ between vectors $X_{t_i}$ and $X_{t_j}$ of time step $t_i$ and $t_j$ for all possible timesteps $t_i, t_j = 1 \dots T$. The resulting matrix $D$ with $D_{ij} = d(X_{t_i}, X_{t_j})$ is often referred to as distance matrix or dissimilarity matrix. The following algorithms are based on pairwise distances.

k-nearest neighbours (*KNN*) can be used for anomaly detection by considering the mean distance to the k-nearest neighbors (k-nearest neighbours Gamma (*KNN-Gamma*)) and the mean length of the vectors pointing from $X_{t_i}$ to its k-nearest neighbors (k-nearest neighbours Delta (*KNN-Delta*)) (Harmeling et al., 2006; Ramaswamy et al., 2000). We fix the hyperparameter $k$ at 10 after carefully trying different choices for $k$ without seeing major effects on preliminary results. Furthermore, we take advantage of the temporal structure of anomalous events excluding 5 neighbouring time steps ($abs(t_i - t_j) \geq 5$) to be also

nearest neighbours.

    Recurrences (*REC*). Within the framework of the theory of nonlinear dynamical systems, each state of a dynamical system will revisit a particular region in its phase space, if waiting for a sufficiently long time (Poincaré, 1890). These dynamics can be visualized in the recurrence plot and are quantified with several metrics usually referred to as recurrence quantification analysis (Marwan et al., 2007). It seems straightforward to use the concept of recurrence analysis to detect states in a dynamical

system that are considered to be rare or unusual. Faranda and Vaienti (2013) used the concept of recurrences and combined it with extreme value theory. We want to use a more general approach without binning the time series. We count the number of observations $\zeta$ falling into a certain $\varepsilon$-ball in a system of multiple variables, condensed by their distance $d(X_{t_i}, X_{t_j})$:

$$\zeta(X_{t_i}) = \sum_{j=1}^{T} \Phi(\varepsilon - d(X_{t_i}, X_{t_j})) \tag{3}$$

$\Phi(z)$ is the Heaviside function, coding the distances to binary values ($\Phi(z) = 0$ if $z < 0$, $\Phi(z) = 1$ otherwise). A $\varepsilon$-hyperball

containing only few recurrent observations is considered to be rare in comparison to the majority of $\zeta$. We compute $1 - \zeta cdot T^{-1}$ to get anomaly scores, which are more likely to be an anomaly for high score values. $\zeta \cdot T^{-1}$ known as local recurrence rate or degree of centrality in recurrence analysis (Marwan et al., 2007; Donner et al., 2010). $\varepsilon$ is the crucial hyperparameter, defining the radius of the ball. Typical choices in recurrence analysis are using quantiles of the distance



matrix, e.g., 5 % or 10 % (Donges et al., 2011a; Flach et al., 2016). As we are not interested in small scale variations falling of *REC*, but more in major anomalies we estimate $\varepsilon$ as median of the distance matrices on the random subsample. This choice turned out to be the optimal choice for $\varepsilon$ in a small simulation, varying the thresholds between the 5 % to 95 % quantile of the distance matrix (not shown). We exclude 5 neighbouring timesteps to be counted as recurrences (similar to *KNN*). *KNN* has

similarities to *REC*, as one could also choose a data-adaptive $k$ such that $\zeta = k$.

The distance matrix $D$ can be transformed into a kernel matrix $K = exp(-0.5 \cdot D \cdot \sigma^{-2})$, i.e. by computing pairwise dissimilarities using Gaussian kernels centered on each data point.

Kernel Density Estimation (*KDE*) is a standard technique for estimating densities based on column means of the kernel matrix $K$ (Parzen, 1962). The bandwidth $\sigma$ of the kernel is a hyperparameter. We estimate $\sigma$ by using the median of the

temporal distance matrix on the random subsample, which is a common choice (Schölkopf and Smola, 2001; Schölkopf et al., 2015).

Support Vector Data Description (*SVDD*) models the distribution of the training data with an enclosing hypersphere in a high-dimensional kernel feature space (Tax and Duin, 2004). As usual a kernel matrix of the random subsample is used for training. Although being a rather simple data description, a hypersphere in the kernel feature space can result in complex

nonlinear decision boundaries in the original space of predictor variables if a nonlinear kernel function is used. Beside the $\sigma$ hyperparameter of the kernel function (see *KDE*), the *SVDD* approach has a parameter called outlier ratio $\nu$ (fixed to 0.2). $\nu$ controls the amount of training samples that can be located outside of the hypersphere to prevent overfitting. As anomaly score for testing, its distance to the center of the hypersphere in the kernel feature space is computed. Testing requires pairwise similarities between test and training samples. For performance reasons in terms of computation time, we used the LIBSVM

(Chang and Lin, 2013) implementation of one-class Support vector machine (Schölkopf et al., 2001), which is an alternative formulation that leads to identical data descriptions as *SVDD* in our setup.

Kernel Null Foley-Sammon Transform (*KNFST*) maps the training data into a so-called null space, in which the training samples have zero variance, i.e., all training samples are mapped to the same point called the target value (Bodesheim et al., 2013). Nonlinearity is incorporated by using a kernel matrix containing pairwise similarities of the training samples (training on

the random subsample as for *SVDD*). Since all training samples are represented by a single target value in the one-dimensional null space, the anomaly score of a test sample is the absolute difference between its projection in the null space and this target value. The projection of the test sample requires pairwise similarities to the training samples. Compared to *SVDD* no parameters need to be tuned except for $\sigma$ of the kernel function that are fixed to the same values for all kernel methods.

### 3.3    Ranking of the Workflows

Given the large number of potential combinations of feature extraction and anomaly detection algorithms, we need an objective criterion to compare the performances of the numerous possible workflows. We use the Area Under the receiver operator characteristics Curve (*AUC*) as our measure of detection skill for a specific event type (Fawcett, 2006). The *AUC* is based on the fraction of events that are correctly detected (true positives) and the fraction of (false) detections among all non-events (false positives), for all possible decision thresholds that could be applied to scores produced by the algorithms. *AUC* values of





0.5 would be achieved by random detection, and values below 0.5 indicate that a lower score is more likely assigned to (true) anomalies than to non-anomalies.

For each data cube with a given event magnitude and event type we compute the *AUC* for each complication, feature extraction and algorithm combination. This leads to an entire catalogue of possible combinations, namely $1.27 \cdot 10^5$ (4 event types, 20 event magnitudes, 11 complications, 18 feature extraction combinations, 8 algorithms). The number of combinations strongly requires simplification to infer knowledge about which combination is advisable to use. Hence, we focus on events of magnitudes typically detected in real world data i.e. changes in the mean (extremes) larger than $2\ sd$ (e.g., temperature extremes in Hansen et al., 2012), a relative increase or decrease in the mean annual cycle amplitude of 25 % (which might happen, e.g. in the carbon cycle after combined drought and heatwaves (Ciais et al., 2005), or in the Arctic due to abrupt sea ice losses (Bintanja and van der Linden, 2013; Bathiany et al., 2016)) or an increase in the signal variance of 25 % (e.g. in temperature, Huntingford et al., 2013).

One way of summarizing the results of such a large number of combinations is treating the *AUC* values as the outcomes of an experiment in which the different design decisions (e.g., feature extraction techniques, anomaly detection algorithms) are the experimental factors. As a control treatment we introduce the simplest possible approach to detect the anomaly: *UNIV* approach on the selected event type, without any further complications (e.g. short extremes or increase measurement noise) on the event type and without prior feature extraction. In order to assess the (averaged) effect of each experimental factor, we fit a linear mixed-effects model (Pinheiro et al., 2016) to the *AUC* data (fixed effects: complications, feature extraction, anomaly detection algorithms; random effect: magnitude of the event). This model's coefficients express the overall effect of a factor level with respect to the control while averaging over all other experimental factors. They are considered to be significant for $p < 0.01$.

Additionally, we compute the Resampling Variation of Parameter estimation of the anomaly detection algorithms (*RVP*) as mean difference of the maximum *AUC* and minimum *AUC* for each resampling $i = 1 \ldots 3$ (Sect. 3.2).

$$RVP_{algorithm} = mean(max(AUC_{comp,feat,magn,event,i}) - min(AUC_{comp,feat,magn,event,i})) \qquad (4)$$

### 3.4 Ensembles of Anomaly Detection Algorithms

Summarizing the output of several anomaly detection algorithms is one way to create more robust results (Thompson, 1977). For better comparability of the algorithms' outputs, we rank them by computing the percentiles of the algorithm scores. These are then aggregated into ensemble scores by computing the Minimum (min, 'Consensus voting'), the Mean ('Balanced voting') or the Maximum (max, 'Risky voting') of the scores of selected well performing algorithms (e.g., Aggarwal, 2012; Zimek et al., 2013).

## 4 Results & Discussion

In the following, we present the performance of the workflows in subsections corresponding to feature extraction techniques (Sect. 4.1), anomaly detection algorithms (Sect. 4.2), and ensembles of detection algorithms (Sect. 4.3). Specifically, we present





**Figure 4.** *AUC* difference with respect to the *UNIV* control in the experimental factors 'feature extraction' and 'detection algorithm' for the event types (a-d).





the *AUC* difference to the *UNIV* control, i.e. the output of the linear mixed-effects model on the experimental factors 'feature extraction' and 'detection algorithm' (Fig. 4). The corresponding tables present the estimates as well as the *RVP* (Tab. 1, 2). Apart from the model the full range of *AUC* values with respect to different event magnitudes, complications and event types is presented in Appendix A, Fig. A1.

## 4.1 Feature Extraction Techniques

Feature extraction techniques are often more important than the detection algorithm itself (Fig. 4). However, we find that choosing a suitable feature extraction technique largely depends on the event type of interest. Therefore, the feature extraction techniques are presented for different event types separately.

*BaseShift*. Shifts in the baseline are simulated to mimic extreme events. Increasing the magnitude (in terms of standard deviations) of a *BaseShift* makes it easier to detect the event (Fig. A1). Dimensionality reduction (via *PCA* or *ICA*) is a crucial feature extraction technique step as it derives meaningful uncorrelated subsets of the data (Fig. 4 a). The combination of dimensionality reduction with some temporal smoothing (*EWMA*) does not exhibit better overall performance (Fig. 4 a) as it fails for *ShortExtremes* due to oversmoothing. Nevertheless *EWMA* can improve the detection rate for special cases, i.e. long events (*LongExtremes*) and high signal to noise ratios (*NoiseIncrease*) (Fig. A1).

*TrendOnset*. Results look very similar to those of *BaseShift*, except that temporal smoothing with *EWMA* has a stronger positive effect than for *BaseShift*. This may be related to the fact that events for *TrendOnset* are longer than those for *BaseShift*. Since the algorithms used in this work are not specifically designed to detect the onset of linear trends, we speculate that their capability to detect such anomalies may be related to their ability to detect base shifts. While algorithms specifically designed to detect changes in trends (e.g., Forkel et al., 2013)) were not included in our work due to our focus on more generic types of anomalies, such specialized algorithms may perform better for this particular class of anomaly.

*MSCChange*. In the detection of *MSCChange* most feature extraction algorithms showed some skill in the detection of an amplitude increase, while only a subset of these succeeded also in detecting decreases in amplitude (Fig. A1). We focus on the the latter ones, which have one step in common: they subtract the median seasonal cycle before applying the detection algorithm (*sMSC*) (Fig. 4 c). In line with the results for *TrendOnset* and *BaseShift*, temporal smoothing in combination with dimensionality reduction improves detection by a large margin (*PCA_sMSC_EWMA*). Furthermore accounting for temporal dynamics with a time delay embedding *TDE* is even more suitable (*TDE_PCA_sMSC_EWMA*).

*VarianceChange*. The algorithms used are hardly able to detect any decrease in variance (Fig. A1). This may be due to an 'overwriting' of the decrease in signal variance with the independent noise since we are using a signal to noise ratio of 0.3. Thus, we exclude a decrease in the variance from the evaluation of the detection algorithms compared to the control. The detection of an increase in the variance can be improved by a combination of dimensionality reduction and variance in a moving window (*PCA_mwVAR*) (Fig. 4 d). Using the variance in a moving window is a popular approach (Huntingford et al., 2013) although it has to be applied with care when used in conjunction with normalization procedures (Sippel et al., 2015).

*SeasonalCycle*. Seasonality is occurring in most EOs. Not accounting for the seasonal cycle has a negative impact on the *AUC* (Appendix A, Fig. A2 a,b,d). However, if we subtract the median cycle within the feature extraction step (*PCA_sMSC_EWMA*,



Fig. 4 a,b,d)), we can almost account for the negative *AUC* impact of the seasonal cycle, as in our experimental setting anomalous events do occur independently of seasonality. However, depending on the research question, independence of seasonality might not always be the case: some EOs may depend, e.g. on vegetation activity, which results in a strong dependence on seasonality.

## 4.2 Performance of Multivariate Anomaly Detection Algorithms

In contrast to the investigated combinations of feature extraction methods, we can identify 3 of the tested algorithms performing on average almost equally well for most event types given a suitable feature extraction as discussed before (Sect. 4.1).

**Table 1.** Average *AUC* difference of the Anomaly Detection Algorithms to the *UNIV* control for each event type.

|  | KNFST | SVDD | T2 | KNN-Delta | KNN-Gamma | KDE | REC |
|---|---|---|---|---|---|---|---|
| *BaseShift* | -0.017 | -0.069 | 0.013 | 0.006 | 0.032 | 0.024 | 0.024 |
| *TrendOnset* | 0.001 | -0.015 | 0.014 | -0.052 | 0.003 | 0.084 | 0.068 |
| *MSCChange* | -0.023 | -0.072 | -0.023 | -0.019 | 0.007 | 0.039 | 0.029 |
| *VarianceChange* | -0.007 | -0.027 | 0.003 | 0.012 | 0.018 | 0.022 | 0.019 |
| Mean | -0.012 | -0.046 | 0.002 | -0.013 | 0.015 | 0.042 | 0.035 |
| RVP | 0.007 | 0.111 | 0.003 |  |  | 0.000 | 0.001 |

*KDE* and *REC* exhibit overall highest *AUC* and lowest *RVP* (Tab. 1). Their estimated mean differences are rather small, since *REC* can be considered as a binary form of the *KDE*. As *REC* uses a threshold $\varepsilon$ for defining the hyperball of recurrences, the results can exhibit slightly higher *AUC* than *KDE* (not shown). However, with *REC* the caveat is that the parameter $\varepsilon$ is not necessarily optimally chosen.

*KNN.* In most of the cases, *KNN-Gamma* performance is better than the *UNIV* control, but only as good as the *UNIV* control for detecting *TrendOnset*. This may be due to the fact that for *TrendOnset*, the mean distance to the *KNN* does not change, unless considering a very large number of *KNN* or excluding a large fraction of temporally near data points to be within the *KNN*. When excluding *TrendOnset* the mean performance increases to 0.019 which is comparable to *KDE* and *REC*. In contrast, *KNN-Delta* does not yield high *AUC*, probably because we do not construct anomalies in the data cube explicitly with a direction that is accounted for by *KNN-Delta* (length of the mean vectors to its *KNN*). The finding that simple algorithms like *KNN-Gamma* (or *KDE*, *T2*) are very competitive, if not favourable algorithms, goes in line with results of Harmeling et al. (2006); Killourhy and Maxion (2009); Ding et al. (2014) on various data sets.

On average, *KNFST* and *SVDD* perform worse than or equally well as the Univariate control algorithm (*UNIV*). Also the *RVP* is highest among the algorithms (Tab. 1). It has already been reported, that *SVDD* can exhibit remarkable fluctuations in the results for sample sizes smaller than 1000 data points (Ding et al., 2014). However, we use 5000 points for training. Thus, we suggest that the fluctuations are due to the fact that *SVDD* and *KNFST* use a training set that is chosen at random and may itself contain anomalies. In the current setting the size of the training sample (5,000) is rather small compared to the spatiotemporal size of the data cube (750,000), and it does not seem to be sufficient to train these algorithms on the data cube.





Increased sample sizes, however, would heavily increase memory demand and computing time, rendering kernel algorithms computationally inapplicable. Training and testing *SVDD* on each pixel did also not improve the results (not shown). We explicitly do not want to state that these 2 algorithms are generally worse, i.e. they are just not built for these massive amounts of data. *KNFST* and *SVDD* outperform others in very different setting (novelty detection in images) (Bodesheim et al., 2013).

*T2* exhibits good performance for detecting starting trends and shifts in the mean. However, it also exhibits the third largest *RVP* (Tab. 1) indicating that the estimation of the covariance matrix may be sensitive to random variation in the data. Nevertheless, the *RVP* is still far better than for *SVDD*. The robust estimation of the mean and covariance matrix might be a difficult task (Smetek and Bauer, 2007; Rousseeuw and Hubert, 2011) for which rather complex algorithms like the (fast) minimum determinant covariance estimator have been proposed, which are closely related to *T2* (Rousseeuw and Van Driessen,

1990). Furthermore, *T2* assumes a multivariate Gaussian distribution and linear dependencies among the variables. Thus, it is not preferable for the complications *NonLinearDep* and *CorrelatedNoise* unless combined with a nonlinear feature extraction technique like ICA (Fig. A1).

## 4.3   Ensembles

The selection of algorithms for computing the ensemble is a compromise between accurate detection of and diversity amongst

the selected algorithms (Zimek et al., 2013). We select the 4 best algorithms (*4b*, *KDE*, *REC*, *KNN-Gamma*,*T2*) and the 3 best distance-based algorithms (*3d*, *KDE*, *REC*, *KNN-Gamma*) for computing their ensembles. We assume that this choice accounts for accuracy (best algorithms selected) as well as for diversity (different algorithms selected).

    Overall, ensemble building improves the anomaly detection rate. The mean *AUC* of each of the ensemble members (*3d*: +0.030, *4b*: +0.023) is lower than the *AUC* of the ensemble, regardless of whether the maximum or the mean is used for

ensemble aggregation. Minimum aggregation of ensemble members, however, performs worse than the individual ensemble members *REC* and *KDE*. Using the maximum or mean yields consistently higher *AUC* than using the minimum (Tab. 2). The superior performance of the maximum choice compared to the minimum indicates that single algorithms overlook more often anomalous events than raising false alarm. Nevertheless, the maximum has the caveat that even a single algorithm may cause a false alarm (Zimek et al., 2013), e.g. due to a poor parameterization or inadequate assumptions about properties of the data.

Thus, a more 'balanced voting' procedure like the mean is the preferable choice and more stable with respect to possible error sources. Among the mean ensembles, the *3d* or *4b* ensembles perform equally well (0.041 vs. 0.039 ±0.001 overall) (Tab. 2).

## 4.4   Limitations

*High Dimensionality*. The utility of distance-based outlier detection algorithms as used in this paper is often questioned in the context of high dimensional data (Zimek et al., 2012). The 'Curse of Dimensionality' states that the difference between

near and far distances diminishes with increasing dimensionality. However, Zimek et al. (2012) showed in the case of *KNN* that the contrary is true for outliers with fixed magnitude in otherwise uncorrelated data. Dimensionality reduction as crucial feature extraction transforms the data into few (ideally) meaningful and uncorrelated variables. Thus, the findings of Zimek





**Table 2.** *AUC* difference of the ensembles of anomaly detection algorithms to the *UNIV* control. Ensembles are computed out of the 4 best algorithms (*4b*, *KDE*, *REC*, *KNN-Gamma*, *T2*) and the 3 best distance-based algorithms (*3d*, *KDE*, *REC*, *KNN-Gamma*).

|  | 3d-max | 3d-mean | 3d-min | 4b-max | 4b-mean | 4b-min |
|---|---|---|---|---|---|---|
| *BaseShift* | 0.042 | 0.037 | 0.033 | 0.042 | 0.038 | 0.030 |
| *TrendOnset* | 0.059 | 0.058 | 0.033 | 0.060 | 0.056 | 0.020 |
| *MSCChange* | 0.033 | 0.040 | 0.032 | 0.033 | 0.037 | 0.017 |
| *VarianceChange* | 0.027 | 0.027 | 0.025 | 0.023 | 0.026 | 0.022 |
| Mean | 0.040 | 0.041 | 0.031 | 0.039 | 0.039 | 0.022 |
| *RVP* | 0.001 | 0.001 | 0.001 | 0.001 | 0.001 | 0.001 |

et al. (2012) provide strong arguments for applying dimensionality reduction on correlated data. We anticipate that his findings are the reason of the superior performance of dimensionality reduction here.

*Heuristic Choices.* Within the parameterization process, several heuristic choices are made. We exclude 5 time steps to be counted as recurrences or k-nearest neighbours. We fix several parameters, e.g. the number of nearest neighbours is fixed to 10. Also other parameter choices are rather heuristic (e.g. $\sigma$), although commonly used. Within the data farm creation, we assume that the number of intrinsic dimension is 3, the signal to noise ratio is 0.3. Furthermore, the choice of the complications might influence the results for each event type, as the standard deviation of *AUC* values over all complications (0.05) is rather large, compared to the average *AUC* gain of the 3 best algorithms with respect to the control (+0.03). However, the ordering of the algorithms is also important to derive rankings of algorithms (Hornik and Meyer, 2007). By choosing different subsets of the complications, we observe that the 3 best algorithms (*KDE*, *REC*, *KNN-Gamma*) are on top, independently of the chosen complication. Therefore, the complications might have an influence on the *AUC* values themselves, but not on the choice of the 3 top candidates.

# 5    Remarks on Applications for Real Earth Observations

Our versions of the artificial data cubes were generated to test different algorithms for their capability to deal with typical properties of Earth observation data. The workflows were chosen to be as generic as possible, and therefore their application to 'real' data with slightly different properties should be made as easy as possible. Nevertheless, several points have to be considered, when applying the algorithms on real EOs.

A typical preprocessing of Earth observations is to center variables to zero-mean and standardize to unit variance (also known as $z$ transformation). A standardization of this kind is of key importance in global EOs. Real multivariate observations often have different physical units or ranges, which have to be made comparable before analysing. However, standardization has to be applied with care. Differences of the mean and variance between geographically distinct or even adjacent grid cells as well as seasonal cycles might corrupt any further analysis. We recommend to subtract the median seasonal cycle before standardization. The median is prefered over the mean as mean seasonal cycles are affected by changes in the amplitude of



the cycle. Standardization can be applied globally (i.e. with global spatiotemporal mean and variance), regionally (i.e. with spatiotemporal mean and variance in subregions of the globe), or locally (i.e. with temporal mean and variance in each grid-cell). Global standardization might be more robust than local, but detects only anomalies in high-variance regions. Local standardization assumes that the number of extreme anomalies is equal in each grid cell, which is a rather strong assumption.

Thus, a regional standardization is favourable in regions with similar mean and variance.

Especially variables presenting a signal from the biosphere are known to exhibit heteroscedasticity, e.g. the variance during growing season is substantially larger than during the rest of the year (Fig. 5). Atmospheric variables in high latitudes also show higher variability during the cold season, e.g. temperature variability might be higher over ice (cold season) than over open water (warm season) (Hansen et al., 2012). Specifically for global applications, using estimates of variance or standard

deviation locally (in each grid-cell) leads to an underestimation of the variance during growing season and thus to an overestimation of anomalies due to standardization especially in the Northern latitudes (Guanche et al., 2016). Thus, we recommend to account for the heteroscedastic pattern by adjusting the variance during the growing season within similar regions. We also recommend this kind of adjustment for the covariance matrix used, e.g., in *T2* or *PCA* as well as for the parameterization of *KDE* or *REC*.

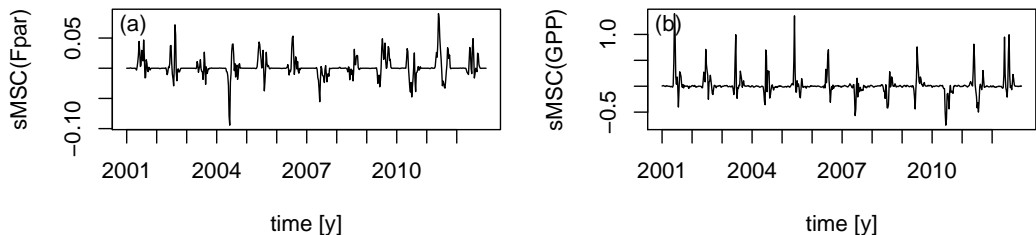

**Figure 5.** The residual time series obtained by subtracting the median seasonal cycle from (a) the fraction of absorbed photosynthetic radiation (fPAR) and (b) Gross Primary Productivity (GPP) at Northern latitudes exhibit heteroscedastic patterns.

Furthermore, anomalies are also overestimated when using a reference period for the estimation of the variance (Sippel et al., 2015). However, with 300 observations in 8-day intervals, as used in this study, this issue is expected to be less pronounced than for fewer observations as it scales with the length of the time series. Nevertheless, we rather recommend to use estimates of the variance of the entire time series or to correct for the overestimation in the out-of-reference period as shown in Sippel et al. (2015).

Regarding the parameterization process of the algorithms we use fixed parameters for $\sigma$, $\varepsilon$, $k$, $\nu$, mean vector, and covariance matrix globally on the entire artificial data cube. Local parameterization assumes the same amount of anomalies in each region, which is neither suitable for the artificial data by construction nor for global real data. Thus we recommend to parametrise globally or within similar regions. Classification of the Earth into similar regions and applying multivariate extreme detection in each region will be the subject of future research.





## 6   Conclusions

Our aim is to identify suitable methods for detecting anomalies in highly multivariate, correlated, and seasonally varying data streams as they are common in Earth system science. In particular, we are interested in detecting shifts in mean (extremes), changes in the amplitude of the seasonal cycle, temporal changes in the variance and onsets of trends. We test a wide range

of workflows (i.e. combining feature extraction techniques and anomaly detection algorithms). All experiments are based on artificial data, designed to mimic real world Earth observations.

We can show that, on average over different anomaly types and data 'complications', 3 multivariate anomaly detection algorithms (*KDE*, *REC*, *KNN-Gamma*) outperform univariate extreme event detection as well as other multivariate approaches (mean *AUC* compared to univariate control: $+0.030$). Additional slight improvement can be achieved by combining the best

algorithms into ensembles using an aggregation by averaging score quantiles ($+0.041$). In contrast, the tested machine-learning algorithms (*SVDD* $-0.05$, *KNFST* $-0.01$) may fail due to overfitting to the training sample.

However, we also find that including a suitable feature extraction technique in the detection workflow is often more important than the choice of the event detection algorithm itself. Yet, we find that the feature extraction has to be explicitly designed for the event type of interest, i.e. time delay embedding (for detecting changes in the cycle amplitude) and exponential weighted

moving average (for detecting trends, long extremes and removing uncorrelated noise in the signal) increases the detection rate of the anomalous events. Including features of the variance within a moving window works partly for detecting increases in the variance, but fails to detect a decrease in the variance due to the relatively high observational noise level. In general, if the data comprises seasonality, subtracting it and using the remaining time series as input feature is essential. Furthermore, we improve the detection rate of multivariate anomalies in highly correlated data streams by adding a dimensionality reduction method to

the workflow (in line with results of Zimek et al., 2012).

The proposed workflows are capable of dealing with common properties of Earth observations like seasonality, non-linear dependencies as well as (to a certain degree) non-Gaussian distributions and noise. Nevertheless, they have to be applied with care to Earth observations, i.e. standardization issues along with strong heteroscedastic patterns (e.g. in Biosphere variables of Northern latitudes) may lead to an overestimation of anomalies. Future work will explore the potential of the identified

workflows on rediscovering known and potentially unknown extremes as well as other anomalies in a set of real Earth system science data streams. We anticipate that an automated application of our workflows might enable the establishment of automated Earth system process control in a very generic manner.





## Appendix A: Detailed Results

**Figure A1.** *AUC* versus event magnitude for all combinations (grey) and the Univariate control (red). Columns of the matrix represent different event types, rows represent complications. Additional colored workflows represent the workflows with the 5 highest mean values for the magnitudes > 2 sd (> 0.6 respectively).



**Figure A2.** Effect of the complications on the 3 best detection algorithms (*KDE*, *REC*, *KNN-Gamma*) presented as *AUC* difference of the *UNIV* control for the event types (a-d).





## Appendix B: Technical Details on Generating the Artificial Data

Within the generation process, we assume that the signal $S$ is additive to the baseline $B$. The baseline might represent reoccuring patterns like seasonality or a constant mean. In addition, binary event parameters $ev_{t,lat,lon}$ are introduced, which allow for switching the anomaly on $ev_{t,lat,lon} \neq 0$ and off $ev_{t,lat,lon} = 0$ ('normality'). The event type and magnitude of the event

is controlled by a parameter separately for the baseline ($k_b$), the signal ($k_s$) and a mean-shift parameter ($k_m$) scaled with the standard deviation of the data $sd$.

$$\Theta_{t,lat,lon} = B_{t,lat,lon} \cdot 2^{(k_b \cdot ev_{t,lat,lon})} + S_{t,lat,lon} \cdot 2^{(k_s \cdot ev_{t,lat,lon})} + k_m \cdot ev_{t,lat,lon} \cdot sd \tag{B1}$$

For a basic version, 3 independent components $\Theta_{t,lat,lon,var}$ are created with the signal consisting of Gaussian noise ($sd = 1$). Each component represents intrinsic properties of the Earth system. Furthermore, we assume that properties of the Earth

system $\Theta_{t,lat,lon}$ are not measured directly but indirectly via a set of correlated variables, i.e. representing patterns of these intrinsic properties. Hence, these variables propagate anomalous events that occur in one independent component. This set of correlated variables $X_{var}$ is created by weighting the intrinsic properties $\Theta_{var}$ with randomly drawn linear (or non-linear) weights $w_j$ plus additional measurement noise $\epsilon$ (Gaussian, $sd = 0.3$) added to each variable.

$$X_{var} = \sum_{j=1}^{j=3} w_j \cdot \Theta_j + \epsilon \tag{B2}$$

Using this data generation scheme, a standard data cube $X_{t_{i,j},lat,lon,var}$ is created, encompassing 300 time steps ($T$), 10 temporally correlated variables ($VAR$) and the total number of latitudes ($LAT$) and longitudes ($LON$) fixed to 50 each. We induce anomalous events with a spatial extent of 40 % of the latitude and longitude and 10 events, each with a temporal extent of 5 time steps. Our total amount of anomalies equals about 3 % of the total data cube.

In the basic version we create 4 data cubes each with a different temporary event type:

– Shift in the baseline, i.e. shift of the running mean of a time series (*BaseShift*) (Fig. 2 a)

   – Change in the variance of the time series (*VarianceChange*) (Fig. 2 b)

   – Change in the amplitude of the mean seasonal cycle of a time series (*MSCChange*)(Fig. 2 c)

   – Onset of a trend in the time series (*TrendOnset*) (Fig. 2 d)

Regarding the complications, some of the event type complication combinations are excluded (Tab. B1). In detail, we do

not expect a *TrendOnset* to 'infect' neighboured cells (*TrendOnset* plus *RandomWalkExtreme*) and a *TrendOnset* can hardly be called a *TrendOnset* if it encompasses only one time step (*ShortExtremes*).

The artificial data farm can be created after cloning into https://github.com/CAB-LAB/DataFarm. Generation is done with the following command within the programming language julia, version 0.4:

*using SurrogateCube; SurrogateCube.DataFarm.makeDataFarm(300,50,50,PathToFolder)*



**Table B1.** Parameter settings for the generation of the artificial data farm. Details are given for each event type and complication (in brackets).

|  |  | Basic | (Complication) | *BaseShift* | *VarianceChange* | *MSCChange* | *TrendOnset* |
|---|---|---|---|---|---|---|---|
| Independent comp. | Θ | 3 | (*MoreIndep Components*) | 3 (6) | 3 (6) | 3 (6) | 3 (6) |
| Dependency (Θ) |  | linear (w) | (*NonLinearDep* squ) | w (squ) | w (squ) | w (sq) | w (sq) |
| Baseline | B | const. = c | (*SeasonalCycle* s, *LatitudinalGradient* lg)) | c (s, lg) | c (s, lg) | s (lg) | c (s, lg) |
| Signal | S | gaussian g | (*LaplacianNoise* l, *CorrelatedNoise* r) | g (l, r) | g (l, r) | g (l, r) | g (l, r) |
| Variables | VAR | 10 |  | 10 | 10 | 10 | 10 |
| Noise | ϵ | 0.3 | (*NoiseIncrease*) | 0.3 (1) | 0.3 (1) | 0.3 (1) | 0.3 (1) |
| Events |  |  |  |  |  |  |  |
| Event number |  | 10 | (*ShortExtremes*, *LongExtremes*) | 10 (50, 5) | 10 (50, 5) | 1 | 1 |
| Spatial extent |  | 1000 |  | 1000 | 1000 | 4 | 1000 |
| Temporal extent |  | 5 | (*ShortExtremes*, *LongExtremes*) | 5 (1,10) | 5 (1,10) | 92 (46, 184) | 150 |
| Magnitudes |  |  |  | $k_m = 0.2\text{-}4$ | $k_s = -2{:}2$ | $k_b = -2{:}2$ | $k_m = 0.2\text{-}4$ |
| Shape |  | rect. | (*RandomWalk Extreme* rw) | rect (rw) | rect (rw) | rect (rw) | rect |

*Author contributions.* M.F. and M.D.M designed the study in collaboration with F.G., A.B., J.D., M.R. and E.R.; M.F. implemented the algorithms including contributions from F.G., P.B. and E.R.; M.F. wrote the manuscript with contributions from all co-authors.

*Acknowledgements.* This research has received funding by the International Max Planck Research School for Global Biogeochemical Cycles (IMPRS), the European Space Agency via the STSE project CAB-LAB and the BACI project, a European Union's Horizon 2020 research and innovation programme under grant agreement No 64176. We thank Simone Girst for her kind language check.



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
