# Peer review of "Multivariate Anomaly Detection for Earth Observations: A Comparison of Algorithms and Feature Extraction Techniques"

_Earth System Dynamics, 2016_

## Referee Comment (RC1) · Anonymous Referee #1 · 9 Dec 2016

\* General comments

The manuscript describes a systematic and comprehensive study of methods for extraction of anomalies and features from artificially-generated multivariate datasets. The presentation is clear, the manuscript is well written, and the study is sound as a comparison of methods for multivariate data analysis, though its value for earth observations is in my opinion not convincing.

Although I understand the rationale for using artificial data, particularly when comparing the performance of different methodological approaches, the artificial events that are considered in the study seem to be unrealistically exaggerated, particularly the amplitude change in the seasonal cycle (Fig. 2 c) and and the change in variance (Fig.

2 d). For example climate related changes in the seasonal cycle or in variance are far more subtle (in terms of magnitude) and much more difficult to identify in real data that the ones exemplified in Fig. 2.

I'm uncomfortable with the term "Complication" used throughout the manuscript to refer to specific characteristics of the artificial data. For example a seasonal cycle can hardly be seen as a complication, it's a feature of the data, not necessarily something complex as it is implicit from denoting it a "complication".

I think that the comprehensiveness of the study is a strength and paradoxically maybe the greatest weakness of the work, because the results need to be necessarily presented in a highly summarized way, here as difference in AUC values (which itself are already a reduction of a ROC curve to a single number) to a univariate approach without "complications" (UNIV). I don't doubt the technical correctness of the results, but in my opinion it's difficult to assess their relevance, particularly in the context of real earth observation data. I find the conclusions of the study quite obvious and realistic (the importance of deseasoning or dimensionality reduction), whether they would require such a wide statistical study on a artificial data farm is not obvious to me.

\* Specific comments

If I understood correctly the length of the generated time series is only of 300 time steps (appendix B), which may be in itself a major factor influencing the performance of some of the methods.

Although I'm keen on the transference of methodological approaches across different areas, and in this case the use of statistical process control (SPC) methods typically used in other contexts (e.g. industry), the restriction of feature extraction methods to the ones used in classical multivariate SPC seems to me an unnecessary restriction. Many feature extraction methods, e.g. wavelets, are routinely used with earth observations precisely because they perform very well in that kind of data.

* Technical corrections

Page 9, line 31: cdot notation

---

## Referee Comment (RC2) · R. V. Donner (Referee) · 7 Feb 2017

Flach et al. present a detailed inter-comparison between a selection of recently applied methodological approaches for detecting multivariate anomalies in Earth observation (EO) data, including a performance assessment based on artificially generated time series data that capture some of the essential features (and complications) of real-world observation. The topic is timely and important, since with the fastly growing amount of big data from remote sensing, the automated identification of key features and particularly unexpected behaviors becomes a crucial task. In this regard, I warmly welcome this study and believe that it can be an important milestone in its field, even though it necessarily presents just a case study and can thus not be complete by

definition.

Prior to accepting this very interesting work for final publication in Earth System Dynamics, I would like to ask the authors to address a couple of questions I came up with when working through their material.

1. It would be good if the authors could clarify already in the abstract which kind of anomalies they aim to address. From Figure 2, it is evident that the four considered types of "events" (or better, episodic behaviors) – base shift, trend onset, change in mean seasonal cycle amplitude, change in variance – affect predominantly the basic statistical features of the data, while their dynamical characteristics (respectively, those of the residuals after removing the seasonal variability component) are largely unaffected. Since this is in contrast to some recent works (including papers by the reviewer's group) which have particularly focused on "dynamical anomalies", it might be worth clarifying this from the beginning. In this context, it is interesting that the authors also consider recurrence characteristics, which are commonly used for detecting changes in the dynamical patterns. However, what they consider here is just a variant of the recurrence rate, which is essentially a statistical characteristic again, as opposed to more sophisticated complexity measures that can be defined within this framework as well.

2. The motivation for choosing the specific settings in the artificial data could be further clarified, especially regarding explicit statements on typical features of real-world EO data. In this context, I was wondering why the authors study only short-term correlated noises, whereas much of the stochastic background signals in common geophysical variables exhibits long-term memory, which might strongly complicate the anomaly detection. Do the authors consider their white/red noises mostly reflecting additive measurement uncertainties or "true" dynamical components, e.g., due to variables and/or scales not resolved by the measurement process.

3. To me, the idea behind mwVAR is not fully clear. Subtracting the median mwVAR
just removes a constant factor from the time series as it is described now. Maybe it should be better explained here how this specific "preprocessing" step works.

4. On p.8, l.22, the authors address the model parameters. However, these parameters have not been introduced before, so it is hard to grasp their meaning at this point.

5. Quite a bit of potentially interesting material is "not shown" by the authors. I understand and agree the need to focus on the most important aspects, but maybe the authors could consider preparing some supplementary material containing these additional results.

6. In general, the parameter selection in the different methods is not well motivated (e.g., embedding delay and dimension, number of nearest neighbors, outlier ratio). Some more words on these aspects would be helpful. The authors shortly address the subjectivity of parameter selection on p.16, ll.3-6, but do not mention that there are established ways to make (some of) these parameter selections at least a bit more objective. I do not request a detailed discussion on this aspect, but it would be worth mentioning it at least.

Technical comments:

- p.1, keywords: please capitalize the names Mahalanobis, Foley and Sammon

- p.3, ll.1-3: The papers by Donges, Rammig, Zscheischler et al. use only a bivariate form of event coincidence analysis. Since the authors refer her to the "truly multivariate" case, a better reference would be Siegmund et al., Front. Plant Sci., 7, 733, 2016, who introduced a multivariate version of event coincidence analysis.

- p.3, l.21: The term "data cube" should be explicitly defined here – it is intuitively clear (especially in connection with Fig. 1), but especially the spatial component (2d vs. 3d) could differ from what is considered in this paper.

- p.3, ll.28-30: It is not clear if the authors wish to consider "multivariate events" or "compound events" (i.e., such that are anomalous with respect to the marginal feature

distribution of a single variable or the joint feature distribution of a (sub)set of variables.

- p.4, l.15: Why is Appendix B referenced in the paper before Appendix A. I think that changing the order of both Appendices would be more logical.

- p.5, ll.6-9: replace a, b, c, d by (a), (b), (c), (d)

- p.5, l.14: I think that it is not the Earth observations (EOs) that are driven by extrinsic forcings, but the EO variables.

- p.6, l.24: In fact, what you study is the maximization of the rate of correct detections at simultaneous minimization of false detections (this is essentially what the ROC analysis does).

- p.6, l.26: "the anomaly time series becomes the feature then" – maybe the authors should explicitly state here what they "define" (consider) to be meant by a feature.

- Figure 3: Since the authors allow for combining different feature extraction techniques, they should emphasize here that their application might be non-commutative in some cases. For example, TDE must be performed after sMSC, otherwise, the signal would be dominated by seasonality and potentially reflect different features than those one is actually interested in.

- p.7, l.9: "This theoretical consideration does not hold true for high dimensional multivariate data." Do the authors have a reference for this? I am not convinced that this statement is correct in general. In particular, one may refer to multi-channel SSA (mSSA), which essentially combines TDE for multivariate data with PCA. What is the difference between mSSA and "dynamic PCA" mentioned in p.7, ll.18-19?

- p.8, l.2: To my knowledge, there are various variants of ICA, and the one maximizing the negentropy is just one version among several others.

- p.8, l.12: "in the literature"

- p.8, l.21: "we fix the model parameters"

- p.8, ll.22-23: "model parameters... and the models themselves... are estimated" – better use the terms model selection and parameter estimation separately- p.8, ll.24-25: Do the authors mean "more resampling is NOT affordable..."?

- p.8, l.26: "a resampling of 3" – 3 what?

- p.8, l.30: "if one or several of the univariate variables are below or above a certain quantile threshold" – again: do the authors mean marginal quantiles or multivariate quantiles (i.e., multivariate or compound extremes)? Page 9, ll.2-3 suggests that they refer to extremes in the marginals.

- p.9, ll.1-2: The event coincidence analysis the authors refer to here is a bivariate (or, in its extension, multivariate) statistical method. Its relevance in the context of the present work is not clear, since I do not find information that statistical interrelationships between anomalies in different variables are considered here.

- p.9, l.3: Details on the definition of the threshold exceedance score should be given.

- p.9, l.12: I suppose that the authors are using standardized variables; otherwise, defining distances across different variables might not make much sense in the real-world data case. I recommend elaborating a bit more on this aspect.

- p.9, ll.15-16: This formulation should be checked again; for me, the difference between the two measures does not become obvious from the given description.

- p.9, ll.18-19: In how far do the authors really "take advantage" here? Isn't it rather that you wish to exclude trivial information due to autocorrelation in your variables?

- p.9, l.29: "An $\epsilon$-hyperball"

- p.9, l.31: $\zeta \cdot T{-1}$

- p.9, l.32: "degree of centrality" is not the proper network theoretic term (it would be "degree centrality" or just "degree"); however, what the authors consider here is not the degree, but the "degree density" (cf. Donges et al., Phys. Rev. E, 85, 046105, 2012).

- p.9, l.33: !quantiles of the distribution of elements of the distance matrix" (also on p.10, ll.3-4)

- p.10, l.20: "of the one-class support vector machine"

- p.10, l.28: "that is fixed"

- p.11, ll.7-9: Temperature extremes represent strong deviations from the mean rather than "changes in the mean".

- p.14, l.17: Do the authors mean "mean length of the vectors"?

- p.16, l.1: "that these findings" or "that their findings"

- p.17, l.22: "parameterise"

- p.18: It is a bit unusual to write the Conclusions completely in present tense. Maybe you wish to consider using present perfect here?

- p.18, ll.12-13: Maybe it is worth clarifying here again that the results apply for the considered types of anomalies?

- Figure A2: It would be interesting to see these charts detailed for the different detection algorithms (e.g. using different colors for the respective bars). Maybe the authors could add some corresponding figure as supplementary material?

- p.21, l.4: I suggest putting the two equations in brackets.

- The authors should check/revise/complete the following citations: Bintanja and van der Linden (2013), Faranda and Vaienti (2013) [remove publisher], Pfeifer et al. (2011) [capitalization of "Earth"], Pinheiro et al. (2016) [capitalization of "R"], Poincaré (1890) [incomplete reference], Smetek and Bauer (2007), van der Maaten (2009), Webber and Marwan (2015) [page numbers].

---

## Author Comment (AC1) · 3 Mar 2017

In the following we are responding to RC1 by an anonymous referee.

\* General comments

Reviewer: The manuscript describes a systematic and comprehensive study of methods for extraction of anomalies and features from artificially-generated multivariate datasets. The presentation is clear, the manuscript is well written, and the study is sound as a comparison of methods for multivariate data analysis, though its value for earth observations is in my opinion not convincing.

Response: We would like to thank the reviewer for the positive feedback. Regarding the

reviewer's concern about the values for EOs, we consider our study as relevant in this context, because current anomaly or extreme detection in Earth observations is mostly done with peak-over-threshold techniques (p. 2, l. 22). These do not consider the multivariate and potentially non-linear correlation structure between multiple variables which we have in EOs. It is therefore an important motivation for our paper to provide a sound basis for alternative and more general approaches. This paper systematically analyses and proposes several algorithms and workflows which consider the structure among multiple variables and furthermore might also reveal novelties about so called compound extremes beyond known patterns, i.e. anomalies where none of the single variables is extreme itself, but their combination is anomalous and leading to an extreme impact. The consideration of compound events does play an increasing role within the community, but is typically confined to known compound events (e.g. heat and droughts) and not very generic. The comparison is performed on artificial data, which were explicitly built to mimic current EOs as ground truth is missing for 'real' EOs (p.3, l.25). Furthermore, available time periods as well as sample sizes are rather small for detecting anomalies in 'real' EOs, which empowers the use of an ensemble of artificial data for method comparison. The application of the proposed workflows to EOs will follow soon.

Reviewer: Although I understand the rationale for using artificial data, particularly when comparing the performance of different methodological approaches, the artificial events that are considered in the study seem to be unrealistically exaggerated, particularly the amplitude change in the seasonal cycle (Fig. 2 c) and and the change in variance (Fig. C1 2 d). For example climate related changes in the seasonal cycle or in variance are far more subtle (in terms of magnitude) and much more difficult to identify in real data that the ones exemplified in Fig. 2.

Response: Please consider that Figure 2 is only an illustration. As described in the manuscript (p. 6, l. 9), we analyse each type of anomaly across 20 different magnitudes - from very minor perturbations to entirely exaggerated values. The generic formula B1

shows that we are effectively exploring the full range of perturbations between very subtle changes (Appendix B: k = 0.2) to exceptionally high changes (k = 4.0) as if it was a model parameter sensitivity analysis. In the revised manuscript, we will add an additional sentence for clarification, explain it in the figure caption and show more realistic magnitudes in Figure 2.

Reviewer: I'm uncomfortable with the term "Complication" used throughout the manuscript to refer to specific characteristics of the artificial data. For example a seasonal cycle can hardly be seen as a complication, it's a feature of the data, not necessarily something complex as it is implicit from denoting it a "complication".

Response: We agree with the Reviewer, that the term 'complication' is far away from being optimal. However, 'data features' might be misunderstood with 'feature extraction techniques', which we wanted to prevent. Therefore, we suggest to rephrase 'complications' into 'data properties' in the revised version.

Reviewer: I think that the comprehensiveness of the study is a strength and paradoxically maybe the greatest weakness of the work, because the results need to be necessarily presented in a highly summarized way, here as difference in AUC values (which itself are already a reduction of a ROC curve to a single number) to a univariate approach without "complications" (UNIV). I don't doubt the technical correctness of the results, but in my opinion it's difficult to assess their relevance, particularly in the context of real earth observation data. I find the conclusions of the study quite obvious and realistic (the importance of deseasoning or dimensionality reduction), whether they would require such a wide statistical study on a artificial data farm is not obvious to me.

Response: We thank the reviewer for this comment. It has two components: (1) the presentation of the results, and (2) the overall relevance. (1) Indeed, we decided to highly summarize the results in the main part of the study. However, more detailed results, for instance the effect of different magnitudes on specific event types and com-

plications can still be inferred from the Appendix Fig. A1. (2) The importance of dimensionality reduction as one way to enhance the performance of anomaly detection algorithms (p.18, l.12) has not been shown before to the best of our knowledge, in particular not for EOs. We are convinced that a highly multivariate system like the Earth with seasonality and potentially non-linear dependencies among the variables requires specific workflows like the ones we propose, i.e. the results of our study are relevant in this context. Currently we are working on applying the algorithms on 'real' EOs with very promising results, i.e., results that capture the major known events globally. Our overarching objective is developing workflows to open a path to a series of scientific studies exploring extreme compound events in depth.

* Specific comments

Reviewer: If I understood correctly the length of the generated time series is only of 300 time steps (Appendix B), which may be in itself a major factor influencing the performance of some of the methods.

Response: Indeed, the length of the time series is a factor influencing the performance of the multivariate anomaly detection algorithms. One crucial point of our study is, that Earth observations are typically short. We seek to understand the performance characteristics of various algorithms and feature extraction methods on short time series. Furthermore, please note that Ding et al. (2014) studied this effect in detail, changing the data set size between 50-30000. The only algorithm on which the size of the data set had a remarkable effect was the Support Vector Data Description (SVDD). SVDD performance increased with the size of the data set. However, even the best performance of SVDD was worse than the other algorithms. Therefore, we conclude that the size of the data set is not influencing the results of the top algorithms (KDE, KNN, REC, T2). We include this aspect in a second version of the paper

Reviewer: Although I'm keen on the transference of methodological approaches across different areas, and in this case the use of statistical process control (SPC) methods

typically used in other contexts (e.g. industry), the restriction of feature extraction methods to the ones used in classical multivariate SPC seems to me an unnecessary restriction. Many feature extraction methods, e.g. wavelets, are routinely used with earth observations precisely because they perform very well in that kind of data.

Response: We are aware that the list of feature extraction algorithms as well as the list of anomaly detection algorithms can hardly be complete. We did not restrict the feature extraction methods only to the ones used in classical statistical process control. We also included non-standard ones from process monitoring in industry (e.g., Independent Component Analysis) and of course from univariate extreme extreme event detection (e.g., subtracting the mean seasonal cycle) (p.6, l. 28). We agree with the reviewer that wavelets perform very well on EOs, e.g., for extracting information about dominant frequencies in the data. However, event detection is another task. We are not aware that wavelets improve the detection rate of multivariate anomalous events, but we will consider this as an interesting aspect for future research.

* Technical corrections

Reviewer: Page 9, line 31: cdot notation

Response: We changed it.

References: Ding, X., Li, Y., Belatreche, A., & Maguire, L. P. (2014). An experimental evaluation of novelty detection methods. Neurocomputing, 135(C), 313–327. http://doi.org/10.1016/j.neucom.2013.12.002
* * *

---

## Author Comment (AC2) · 6 Mar 2017

We would like to thank the reviewer for his positive feedback and the numerous suggestions for improvement of the paper. In the following, we respond to each of the reviewer comments in detail.

*Reviewer:*
Flach et al. present a detailed inter-comparison between a selection of recently applied methodological approaches for detecting multivariate anomalies in Earth observation (EO) data, including a performance assessment based on artificially generated time series data that capture some of the essential features (and complications) of realworld observation. The topic is timely and important, since with the fastly growing amount of big data from remote sensing, the automated identification of key features and particularly unexpected behaviors becomes a crucial task. In this regard, I warmly welcome this study and believe that it can be an important milestone in its field, even though it necessarily presents just a case study and can thus not be complete by definition.

Prior to accepting this very interesting work for final publication in Earth System Dynamics, I would like to ask the authors to address a couple of questions I came up with when working through their material.

1. It would be good if the authors could clarify already in the abstract which kind of anomalies they aim to address. From Figure 2, it is evident that the four considered types of "events" (or better, episodic behaviors) – base shift, trend onset, change in mean seasonal cycle amplitude, change in variance – affect predominantly the basic statistical features of the data, while their dynamical characteristics (respectively, those of the residuals after removing the seasonal variability component) are largely unaffected. Since this is in contrast to some recent works (including papers by the reviewer's group) which have particularly focused on "dynamical anomalies", it might be worth clarifying this from the beginning. In this context, it is interesting that the authors also consider recurrence characteristics, which are commonly used for detecting changes in the dynamical patterns. However, what they consider here is just a variant of the recurrence rate, which is essentially a statistical characteristic again, as opposed to more sophisticated complexity measures that can be defined within this framework as well.

*Response:*
We agree with the reviewer that our artificial detection experiment does mostly not affect dynamical characteristics of time series which might be revealed by numerous more complex measures derived from recurrence quantification techniques or recurrence networks. However, the experiment was also not meant to do so. We focus on basic time series characteristics, which are often perceived as "extremes" in the public. As proposed by the reviewer we extend the sentence in the abstract for clarification as follows (p.1, l.9): We rely on artificial data that mimic typical properties and anomalies in multivariate spatiotemporal Earth observations *like sudden changes of basic characteristics of time series such as the sample mean, the variance, changes in the cycle amplitude and trends.*

*Reviewer:*
2. The motivation for choosing the specific settings in the artificial data could be further clarified, especially regarding explicit statements on typical features of real-world EO data. In this context, I was wondering why the authors study only short-term correlated noises, whereas much of the stochastic background signals in common geophysical variables exhibits long-term memory, which might strongly complicate the anomaly detection. Do the authors consider their white/red noises mostly reflecting additive measurement uncertainties or "true" dynamical components, e.g., due to variables and/or scales not resolved by the measurement process.

*Response:*
We thank the reviewer for this comment. According to the reviewer's suggestion, we will motivate the settings of the artificial data with typical real-world EO data features at p.5, l. 5:

1. *Shift in the baseline, i.e. shift of the running mean of a time series (BaseShift) (Fig. 2 (a)). This event type is closely related to "extremes" in real-world Earth observations.*

2. *Onset of a trend in the time series (TrendOnset) (Fig. 2 (b)).*

[Figure]

3. *Change in the amplitude of the mean seasonal cycle of a time series (MSCChange) (Fig. 2 (c)), which might happen in the real-world carbon cycle as response to combined drought-heatwaves (Ciais et al., 2005).*

4. *Change in the variance of the time series (VarianceChange) (Fig. 2 (d)), e.g., in temperature (Huntingford et al., 2013).*

In real-world EO data, we are typically dealing with rather short time series (e.g., less than 10-15 years). Long-term memory processes cannot reliably diagnosed with such short time series (Ghil et al., 2011, e.g.). However, our main focus in this paper is distinguishing anomalies from 'normal' short term noise and not to detect dynamical changes in the processes of a system or to infer anything about the (dynamical) reason behind the anomaly. In case an extreme anomalies occurs, it will certainly impact people, ecosystems, etc., regardless whether it was a random natural event or due to a change in the system's dynamics. In this paper, we want to detect such kind of events. Therefore, we consider the red/white noise in our artificial data farm to reflect both 'true' dynamical components (esp. In the 'signal' of the independent components) as well as measurement uncertainty (which is also explicitly added as additional white noise on the top of each variable). It would indeed be a very different but nevertheless very interesting question how multivariate anomaly detection algorithms perform in the presence of long-term memory and how to distinguish anomalies which occur due to long-term memory from anomalies in short term noise. This question is beyond the scope of this paper, but we thank the reviewer for this interesting aspect for future research.

*Reviewer:*
3. To me, the idea behind mwVAR is not fully clear. Subtracting the median mwVAR just removes a constant factor from the time series as it is described now. Maybe it should be better explained here how this specific "preprocessing" step works.

*Response:*
We thank the reviewer to point to this formulation and understanding issue. Subtracting a constant (the median of moving window variance) is indeed not influencing the results of algorithms based on pairwise distances. We rephrase the paragraph (p.7, l.4-6) to:
*Computing the moving window Variance (mwVAR) is a popular technique for detecting trends in the variance in univariate time series (e.g., Huntingford et al., 2013). We choose a window size of 10 and compute the variance in the running window along the time series of each variable. We use the estimates of the mwVAR time series as feature to detect multivariate anomalies in the variance.*

*Reviewer:*
4. On p.8, l.22, the authors address the model parameters. However, these parameters have not been introduced before, so it is hard to grasp their meaning at this point.

*Response:*
For better understandability, it seems to be more logical to us to describe the parameter estimation procedure once, before introducing the anomaly detection algorithms. To clarify, we changed p.8, l. 22 to: *S*ome anomaly detection algorithms require the estimation of parameters (Details are given below for each algorithm separately). In that case we fix the model parameters for the entire data cube. We estimate model parameters ($\sigma$, $\varepsilon$, $Q$, $\mu$, see below) and train the models themselves (Support Vector Data Description, Kernel Null Foley-Sammon Transform, see below) based on a random subsample of 5000 data points obtained from the entire data cube.

*Reviewer:*
5. Quite a bit of potentially interesting material is "not shown" by the authors. I understand and agree the need to focus on the most important aspects, but maybe the authors could consider preparing some supplementary material containing these additional results.

*Response:*
We are pleased that the reviewer is interested in additional aspects which we did not show. We are only aware of two aspects, which are referred to but not shown in the paper:

1. "Training and testing SVDD on each pixel did also not improve the results" (p.15, l.2). We thank the reviewer for pointing to this aspect as this is not only an experimental result, but even a theoretical finding. Training and testing SVDD with the same parameterisation ($\nu$) on each pixel assumes the same number of anomalous events in each pixel. Therefore, it cannot improve the detection rate in datasets with varying number of anomalous events. We propose to rephrase the sentence (p.15, l.2) to:

   *Training and testing SVDD on each pixel does also not improve the results as the amount of anomalies differs between different pixels in our setting. This contrasts the assumption when training SVDD on each pixel with constant outlier ratio ($\nu$ parameter).*

2. AUC values of different $\sigma$ (KDE) or $\varepsilon$ (REC) choices (p.10, l.4 and p.14, l.10). We will prepare supplementary material including one figure (S1, attached), which shows a small simulation (500 repetitions) in which we are trying to detect one anomalous event (BaseShift) with different $\sigma$ (KDE) or $\varepsilon$ (REC) choices. We change $\sigma$ (or $\varepsilon$, respectively) between the 0.05 and 0.95 quantile of the distance matrix. Results exhibit, that REC has slightly higher AUC values for optimal $\varepsilon$ choices, whereas KDE is largely insensitive to different $\sigma$ choices in the given range.

*Reviewer:*
6. In general, the parameter selection in the different methods is not well motivated

(e.g., embedding delay and dimension, number of nearest neighbours, outlier ratio). Some more words on these aspects would be helpful. The authors shortly address the subjectivity of parameter selection on p.16, ll.3-6, but do not mention that there are established ways to make (some of) these parameter selections at least a bit more objective. I do not request a detailed discussion on this aspect, but it would be worth mentioning it at least.

*Response:*
We explicitly wanted to point to the heuristic parameters choices, as we are aware that this is a crucial aspect for our results. Therefore, we selected the parameters very carefully. However, we assume that the reviewer's concerns especially about the time delay embedding for which much more objective criteria exist and apologize for not mentioning these criteria in the manuscript. To address this issue we will extend the following sentences:

1. p.16, ll.5-6: The artificial data farm's intrinsic dimension is 3 as it was created from three independent components. Therefore the embedding dimension $m$ is fixed accordingly although it can be inferred based on the data by determining the number of false nearest neighbours (Kennel et al., 1992; Hegger et al., 1999).

2. p.7, l.11: We fix $m$ to 3 (*corresponding to the number of independent components within the data farm creation*) and $\tau$ to 6 which is a compromise between the typical choice of the first zero crossing of the temporal autocorrelation function *or the first local minimum of the mutual information* (Fraser and Swinney, 1986; Webber and Marwan, 2015).

**Technical comments:**

*Reviewer:*
- p.1, keywords: please capitalize the names Mahalanobis, Foley and Sammon

*Response:*
Done

*Reviewer:*
- p.3, ll.1-3: The papers by Donges, Rammig, Zscheischler et al. use only a bivariate form of event coincidence analysis. Since the authors refer her to the "truly multivariate" case, a better reference would be Siegmund et al., Front. Plant Sci., 7, 733, 2016, who introduced a multivariate version of event coincidence analysis.

*Response:*
We now refer additionally to Siegmund et al. (2016) at p.3, ll.1-3 as well as on p.9, l.2.

*Reviewer:*
- p.3, l.21: The term "data cube" should be explicitly defined here – it is intuitively clear (especially in connection with Fig. 1), but especially the spatial component (2d vs. 3d) could differ from what is considered in this paper.

*Response:*
We defined it now as follows (p.3, l.21): *S*patio-temporal EOs are therefore stored in the Earth system data cube, which is a 4 dimensional array of latitudes, longitudes, time and different measurement variables. To detect multivariate anomalies in EOs, we define an anomaly to be any consecutive spatiotemporal part of the data cube ...

*Reviewer:*
- p.3, ll.28-30: It is not clear if the authors wish to consider "multivariate events" or "compound events" (i.e., such that are anomalous with respect to the marginal feature distribution of a single variable or the joint feature distribution of a (sub)set of variables.

*Response:*
We thank the reviewer for this question. It is definitely within our scope to consider also "multivariate events", i.e. anomalous events where none of the single variables is extreme itself, but their joint feature distribution is anomalous. However, "compound events" are usually defined as multivariate events which are additionally leading to an extreme impact (Seneviratne et al., 2012; Leonard et al., 2013), which is not possible to evaluate with the artificial data farm. Nevertheless, we consider our study an important scoping study also in the context of compound events, as the proposed algorithms and workflows are in general capable to detect multivariate anomalous events, which might include compound events (with impact) in real EOs. For clarification we add (p.3, l.28-30): Second, we use these artificial data to evaluate the capability of different algorithms to detect multivariate anomalous events, *including compound events (i.e. events where none of the single variables is extreme, but their joint distribution is anomalous and might lead to an extreme impact) (Seneviratne et al., 2012; Leonard et al., 2013).*

*Reviewer:*
- p.4, l.15: Why is Appendix B referenced in the paper before Appendix A. I think that changing the order of both Appendices would be more logical.

*Response:*
We changed the order according to the reviewer's suggestions.

*Reviewer:*
- p.5, ll.6-9: replace a, b, c, d by (a), (b), (c), (d)

*Response:*
Done

*Reviewer:*
- p.5, l.14: I think that it is not the Earth observations (EOs) that are driven by extrinsic forcings, but the EO variables.

*Response:*
We changed EOs to EO variables on p.5, l.14.

*Reviewer:*
- p.6, l.24: In fact, what you study is the maximization of the rate of correct detections at simultaneous minimization of false detections (this is essentially what the ROC analysis does).

*Response:*
We thank the reviewer for this comment. However, we are not convinced that mentioning details on ROC analysis facilitates understanding of the essential point here, which deals with the term feature extraction and its justification. To clarify we change the term "event detection rate" to "detection of anomalous events" (p.6, l.24). The exact definition of ROC/AUC characteristics is given later, p. 10, ll.31-33.

*Reviewer:*
- p.6, l.26: "the anomaly time series becomes the feature then" – maybe the authors should explicitly state here what they "define" (consider) to be meant by a feature.

*Response:*
The definition of feature extraction is already given a few sentences before (p.6, l. 23). Therefore, we changed the sentence (p.6, l. 26) to: *A very simple form of feature extraction could be to subtract the mean seasonal cycle. We consider the anomaly time series to be the extracted feature in this case.*

*Reviewer:*
- Figure 3: Since the authors allow for combining different feature extraction techniques, they should emphasize here that their application might be non-commutative in some cases. For example, TDE must be performed after sMSC, otherwise, the signal would be dominated by seasonality and potentially reflect different features than those one is actually interested in.

*Response:*
We thank the reviewer for this comment and add a sentence on that (p.8, l. 19): *In some cases this might lead to non-commutative combinations, especially for non-linear feature extraction techniques.*

*Reviewer:*
- p.7, l.9: "This theoretical consideration does not hold true for high dimensional multivariate data." Do the authors have a reference for this? I am not convinced that this statement is correct in general. In particular, one may refer to multi-channel SSA (mSSA), which essentially combines TDE for multivariate data with PCA. What is the difference between mSSA and "dynamic PCA" mentioned in p.7, ll.18-19?

*Response:*
We thank the reviewer for his comment on multi-channel SSA. Dynamic PCA and mSSA are not different in technique, although their purpose differs (extracting main frequencies, versus smoothing for subsequent process monitoring). We removed the statement about the theoretical consideration of high dimensional multivariate data.

*Reviewer:*
- p.8, l.2: To my knowledge, there are various variants of ICA, and the one maximizing the negentropy is just one version among several others.

[Figure]

*Response:*
The currently used formulation was indeed not ideal. We specify the sentence as follows: *We use one ICA variant which tries to separate different sources of data by maximizing the negentropy*

*Reviewer:*
- p.8, l.12: "in the literature"

*Response:*

Done

*Reviewer:*
- p.8, l.21: "we fix the model parameters"

*Response:*
Done

*Reviewer:*
- p.8, ll.22-23: "model parameters. . . and the models themselves. . . are estimated" – better use the terms model selection and parameter estimation separately

*Response:*
Done

*Reviewer:*
- p.8, ll.24- 25: Do the authors mean "more resampling is NOT affordable. . ."?

*Response:*
Yes, indeed. We changed it.

*Reviewer:*
- p.8, l.26: "a resampling of 3" – 3 what?

*Response:*
... 3 times. We changed it.

*Reviewer:*
- p.8, l.30: "if one or several of the univariate variables are below or above a certain quantile threshold" – again: do the authors mean marginal quantiles or multivariate quantiles (i.e., multivariate or compound extremes)? Page 9, ll.2-3 suggests that they refer to extremes in the marginals.

*Response:*
For the "univariate approach" we refer to quantiles in the distribution of each single variable separately, i.e. to extremes in the marginals. To clarify we changed p.8, l.30 to: In this case, one would consider a data point to be extreme, if one or several of the univariate variables are above (or below) a certain quantile threshold *of the marginal distributions of each single variable*.

*Reviewer:*
- p.9, ll.1-2: The event coincidence analysis the authors refer to here is a bivariate (or, in its extension, multivariate) statistical method. Its relevance in the context of the present work is not clear, since I do not find information that statistical interrelationships between anomalies in different variables are considered here.

*Response:*
We totally agree with the reviewer that the mentioned coincidence analysis do not consider interrelationships between different variables. Therefore, we also write p.8, ll.29-30, that the technique is "multiple univariate". We would not consider it, to be a "real" multivariate technique as the following ones. However, it is the simplest technique for detecting anomalies in multiple data streams. Thus, we use the technique as benchmarking for the other algorithms.

*Reviewer:*
- p.9, l.3: Details on the definition of the threshold exceedance score should be given.

*Response:*
We will add the details on that: different thresholds in terms of quantiles of the marginal distributions between 0.0 to 1.0 (accuracy 0.01) are used.

*Reviewer:*
- p.9, l.12: I suppose that the authors are using standardized variables; otherwise, defining distances across different variables might not make much sense in the real-world data case. I recommend elaborating a bit more on this aspect.

*Response:*
In our artificial data, the variables are already comparable by construction, so standardisation is not needed. However, for the real-world data standardisation is important. Furthermore it might even be an additional error source, if not applied with care. We elaborate already on that on p. 16, l.18, but nevertheless add an additional sentence for clarification: For real-world data, variables have to be standardized with care before computing the distance matrix (Sect. 5). However, in our artificial data farm the variables are already comparable by construction, thus standardization is not needed.

*Reviewer:*
- p.9, ll.15-16: This formulation should be checked again; for me, the difference between the two measures does not become obvious from the given description.

*Response:*
We change it to: K-nearest neighbours (KNN) can be used for anomaly detection by considering the mean distance to the k-nearest neighbors (KNN-Gamma) and the length of the mean of the vectors pointing from $X_t$ to its k-nearest neighbors (KNN-Delta). With that approach KNN-Delta considers also the direction of the neighbors, i.e. has higher values in case its nearest neighbours are pointing in one direction, which is in contrast to the directionless distance of KNN-Gamma.

*Reviewer:*
- p.9, ll.18-19: In how far do the authors really "take advantage" here? Isn't it rather that you wish to exclude trivial information due to autocorrelation in your variables?

*Response:*
Indeed, "take advantage" was rather meant in the sense of improving the algorithm's capability to deal with autocorrelated data. Thus, we reformulate according to the reviewers suggestion to: *We exclude trivial temporal autocorrelations* by excluding 5 neighbouring time steps to be also nearest neighbours.

*Reviewer:*
- p.9, l.29: "An epsilon-hyperball"

*Response:*
Done

*Reviewer:*
- p.9, l.31: $\zeta \cdot T^{-1}$

*Response:*
Done

*Reviewer:*
- p.9, l.32: "degree of centrality" is not the proper network theoretic term (it would be "degree centrality" or just "degree"); however, what the authors consider here is not the degree, but the "degree density" (cf. Donges et al., Phys. Rev. E, 85, 046105, 2012).

*Response:*
We thank the reviewer pointing this out and changed the term in degree density, as well as adding the correct citation.

*Reviewer:*
- p.9, l.33: "quantiles of the distribution of elements of the distance matrix" (also on p.10, ll.3-4)

*Response:*
Done

*Reviewer:*
- p.10, l.20: "of the one-class support vector machine"

*Response:*
Done

*Reviewer:*
- p.10, l.28: "that is fixed"

*Response:*
Done

*Reviewer:*
- p.11, ll.7-9: Temperature extremes represent strong deviations from the mean rather than "changes in the mean".

*Response:*
We changed "changes in the mean" to "deviations from the mean".

*Reviewer:*
- p.14, l.17: Do the authors mean "mean length of the vectors"?

*Response:*
Indeed, we changed it.

*Reviewer:*
- p.16, l.1: "that these findings" or "that their findings"

*Response:*
Changed to "their findings".

*Reviewer:*
- p.17, l.22: "parameterise"

*Response:*
Done

*Reviewer:*
- p.18: It is a bit unusual to write the Conclusions completely in present tense. Maybe you wish to consider using present perfect here?

*Response:*
We will rewrite the first paragraph of the Conclusions in past tense.

*Reviewer:*
- p.18, ll.12-13: Maybe it is worth clarifying here again that the results apply for the considered types of anomalies?

*Response:*
We added "for the considered event types" for clarification.

*Reviewer:*
- Figure A2: It would be interesting to see these charts detailed for the different detection algorithms (e.g. using different colors for the respective bars). Maybe the authors could add some corresponding figure as supplementary material?

*Response:*
We will prepare an additional figure as suggested by the reviewer.

*Reviewer:*
- p.21, l.4: I suggest putting the two equations in brackets.

*Response:*
Done

*Reviewer:*
- The authors should check/revise/complete the following citations: Bintanja and van der Linden (2013), Faranda and Vaienti (2013) [remove publisher], Pfeifer et al. (2011) [capitalization of "Earth"], Pinheiro et al. (2016) [capitalization of "R"], Poincare (1890) [incomplete reference], Smetek and Bauer (2007), van der Maaten (2009), Webber and Marwan (2015) [page numbers].

*Response:*
Done

**References**

Ciais, P., Reichstein, M., Viovy, N., Granier, A., Ogee, J., Allard, V., Aubi- net, M., Buchmann, N., Bernhofer, C., Carrara, A., Chevallier, F., De No- blet, N., Friend, A. D., Friedlingstein, P., Gruenwald, T., Heinesch, B., Keronen, P., Knohl, A., Krinner, G., Loustau, D., Manca, G., Matteucci, G., Miglietta, F., Ourcival, J. M., Papale, D., Pilegaard, K., Rambal, S., Seufert, G., Soussana, J. F., Sanz, M. J., Schulze, E. D., Vesala, T., and Valentini, R. (2005). Europe-wide reduction in primary productivity caused by the heat and drought in 2003. Nature, 437(7058):52–533.

Fraser, A. M. and Swinney, H. L. (1986). Independent coordinates for strange attractors from mutual information. Physical Review A, 33:1–7.

Ghil, M., Yiou, P., Hallegatte, S., Malamud, B. D., Naveau, P., Soloviev, A., Friederichs, P., Keilis-Borok, V., Kondrashov, D., Kossobokov, V., Mestre, O., Nicolis, C., Rust, H. W., Shebalin, P., Vrac, M., Witt, A., and Zaliapin, I. (2011). Extreme events: dynamics, statistics and prediction. Nonlinear Processes in Geophysics, 18(3):295–350.

Hegger, R., Kantz, H., and Schreiber, T. (1999). Practical implementation of nonlinear time series

methods: The TISEAN package. Chaos: An Interdisciplinary Journal of Nonlinear Science, 9(2):413.

Huntingford, C., Jones, P. D., Livina, V. N., Lenton, T. M., and Cox, P. M. (2013). No increase in global temperature variability despite changing regional patterns. Nature, 500(7462):327–330.

Kennel, M. B., Brown, R., and Abarbanel, H. D. I. (1992). Determining embedding dimension for phase-space reconstruction using a geometrical construction. Physical Review A, 45:3403–3411.

Leonard, M., Westra, S., Phatak, A., Lambert, M., van den Hurk, B., McInnes, K., Risbey, J., Schuster, S., Jakob, D., and Stafford-Smith, M. (2013). A compound event framework for understanding extreme impacts. Wiley Interdisciplinary Reviews: Climate Change, 5(1):113–128.

Seneviratne, S. I., Nicholls, N., Easterling, D., Goodess, C., Kanae, S., Kossin, J., Luo, Y., Marengo, J., McInnes, K., Rahimi, M., Reichstein, M., Sorteberg, A., Vera, C., and Zhang, X. (2012). Changes in climate extremes and their impacts on the natural physical environment. In Field, C., Barros, V., Stocker, T., Qin, D., Dokken, D., Ebi, K., Mastrandrea, M., Mach, K., Plattner, G.-K., Allen, S., Tignor, M., and Midgley, editors, Managing the Risks of Extreme Events and Disasters to Advance Climate Change Adaptation (IPCC SREX Report), pages 109–230. Cambridge University Press.

Siegmund, J. F., Sanders, T. G. M., Heinrich, I., van der Maaten, E., Simard, S., Helle, G., and Donner, R. V. (2016). Meteorological Drivers of Extremes in Daily Stem Radius Variations of Beech, Oak, and Pine in North-eastern Germany: An Event Coincidence Analysis. Frontiers in Plant Science, 7:220.

Webber, Jr., C. L. and Marwan, N. (2015). Mathematical and Computational Foundations of Recurrence Quantifications. In Recurrence Quantification Analysis, pages 3–43. Springer, Cham Heidelberg New York Dordrecht London.

---

## Author Response (AR2)

Response on the

**Review on "Multivariate Anomaly Detection for Earth Observations: A Comparison of Algorithms and Feature Extraction Techniques" by Milan Flach et al.**
    Reviewer: R. V. Donner (Report # 1, Referee #2), 17. May 2017

*Reviewer:*
The authors have successfully addressed my previous comments. I believe that this manuscript has the potential to become an important reference for future works and thus recommend its publication with the following minor amendments.

*Response:*
We would like to thank the reviewer for this positive evaluation. We addressed all minor technical issues as outlined below.

*Reviewer:*
- minor spell-checking (there are a few typos, particularly regarding use of singular vs. plural forms)

*Response:*
We corrected some minor typos and plural forms. We changed non-linear to nonlinear, neighbors to neighbours for consistency throughout the entire manuscript.

*Reviewer:*
- p.7, l.11: clarify that these "anomalies" (in the climatological sense) have nothing to do with the notion of anomalies used in this paper - should be clear from the context, but better state this explicitly.

*Response:*
We thank the reviewer for this hint and state explicitly now (p.7, ll.10-14): The remaining part of the time series is often referred to as 'anomalies' in the climatological sense. These anomalies are used here as input feature. Please note, that the 'climatological' anomalies are only the difference from the mean behaviour and thus are not to be mixed up with anomalies (strange or rare regions in the data, closely related to extreme events) as detected through the (multivariate) anomaly detection algorithms (Sect. 3.2).

*Reviewer:*
- p.10, l.21: what was the criterion for optimality?

*Response:*
added optimal 'in therms of maximizing the Area Under the Curve' (p.10, l.25)

*Reviewer:*
- when referring to Fig. S1 and S2, please state explicitly that they can be found in the Supplementary Material

*Response:*
Added explicitly 'Supplementary Material' when referring to the Fig. S1. We obviously missed to refer to Fig. S2 in the text and therefore added (p.15, ll.7-9): In addition, we observe even superior performance of *KNN-Gamma* compared to *KDE* and *REC* for 'difficult' data properties (e.g. *MoreIndepComponents, CorrelatedNoise*, Supplementary Materials, Fig. S2).

*Reviewer:*
- cross-check/update the following references: Bintanja and van der Linden 2013 (article ID/pages missing), Guanche et al. 2016 (paper meanwhile published), Hansen et al. 2012 (volume?), Marwan et al. 2007 (2nd author should read Romano, M.C.), Nagendra et al. 2012 (volume?)

*Response:*
We changed the above mentioned references and cross checked also all other citations. Additionally changed:
Hegger et al., 1999, Chaos (page numbers), Fraser & Swinney, 1986, Physical Review A (page numbers), Bae et al., 2003, Review of Financial Studies (corrected name of the coauthor)

**Further Changes:** We added 'compound events' to the keywords.

[revised manuscript text omitted]